# EDGE REWIRING GOES NEURAL: BOOSTING NETWORK RESILIENCE VIA POLICY GRADIENT

## ABSTRACT

Improving the resilience of a network protects the system from natural disasters and malicious attacks. This is typically achieved by introducing new edges, which however may reach beyond the maximum number of connections a node could sustain. Many studies then resort to the degree-preserving operation of rewiring, which swaps existing edges $AC, BD$ to new edges $AB, CD$. A significant line of studies focuses on this technique for theoretical and practical results while leaving three limitations: network utility loss, local optimality, and transductivity. In this paper, we propose **ResiNet**, a reinforcement learning (RL)-based framework to discover **Resi**lient **Net**work topologies against various disasters and attacks. ResiNet is objective agnostic which allows the utility to be balanced by incorporating it into the objective function. The local optimality, typically seen in greedy algorithms, is addressed by casting the cumulative resilience gain into a sequential decision process of step-wise rewiring. The transductivity, which refers to the necessity to run a computationally intensive optimization for each input graph, is lifted by our variant of RL with auto-regressive permutation-invariant variable action space. ResiNet is armed by our technical innovation, **Fil**t**r**ation **e**nhanced GNN (**FireGNN**), which distinguishes graphs with minor differences. It is thus possible for ResiNet to capture local structure changes and adapt its decision among consecutive graphs, which is known to be infeasible for GNN. Extensive experiments demonstrate that with a small number of rewiring operations, ResiNet achieves a near-optimal resilience gain on multiple graphs while balancing the utility, with a large margin compared to existing approaches.

## 1 INTRODUCTION

Modern infrastructure systems, such as computer routing and electric power networks, are vulnerable to natural disasters and malicious attacks (Schneider et al., 2011). Consider the scenario where the abnormality of one power supply station causes other power supply stations to overload, which cascades more power supply to fail, resulting in a region-wide power outage. Figure 1 visualizes this scenario that the failures of a dozen of nodes could jeopardize the connectivity and utility of the EU power network with 217 nodes. The ability for a system to defend itself from such failures and attacks is characterized by the *network resilience*. A resilient network should continue to function and maintain an acceptable level of utility when part of the network fails.

A network becomes more resilient if some connections could backup the others. It is seemingly straightforward to achieve resilience by adding redundant edges. However, it will not be practically feasible as the nodes are usually already running at their full capacity. For example, a power supply facility can only support a certain number of connections and cannot afford additional loads. In such a regime, degree-preserving operations, such as edge rewiring, are desired (Schneider et al., 2011; Rong & Liu, 2018; Yazıcıoğlu et al., 2015; Zhou & Liu, 2014). Let $G = (V, E)$ be a graph. An edge rewiring operation alters the graph structure by removing $AC$ and $BD$ and adding $AB$ and $CD$, where $AC, BD \in E$ and $AB, CD, AD, BC \notin E$.[1] Edge rewiring is empirically effective to maximize network resilience and becomes the prevailing atomic operation in the literature. Despite this, methods for improving network resilience via edge rewiring share several limitations: utility loss, local optimality, and transduction.

---

[1] A few works have a different definition for the edge rewiring operation.

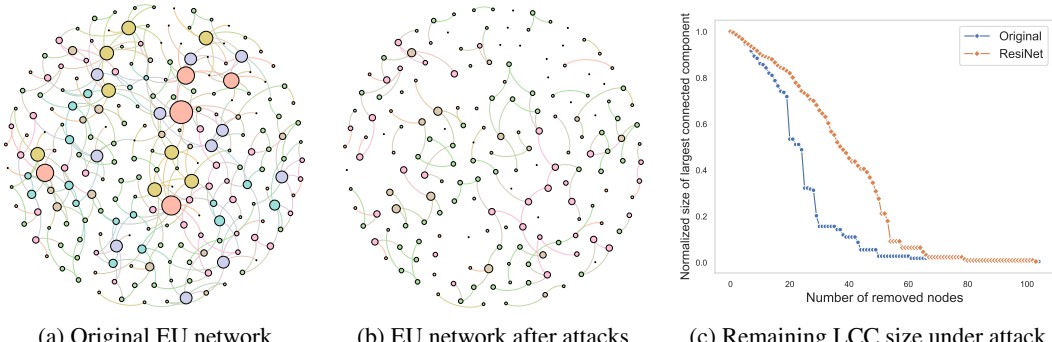

(a) Original EU network          (b) EU network after attacks          (c) Remaining LCC size under attack

Figure 1: The EU power network under the adaptive degree-based attack which removes the most critical node recursively with (a) original EU network with 217 nodes, (b) remaining EU network after a series of attacks on 40 nodes, and (c) the change of the normalized size of the largest connected component (LCC). The node size is proportional to its degree and the node color is given by DBSCAN (Ester et al., 1996).

- Network utility loss. Improving network resilience aims to protect the network utility from degrading under attack and make the network continue functioning. Optimizing network resilience without considering the network utility could jeopardize the functioning without the presence of an attack.

- Local optimality. Combinatorially choosing the edges to rewire to optimize the resilience is proved to be NP-hard (Mosk-Aoyama, 2008). Previous studies predominantly seek approximate optimality through greedy-like algorithms, which yields local optimality in practice.

- Transductivity. Traditional resilience optimization methods are transductive since they search the resilience topology on a given graph. This search process has to be executed for every graph and does not transfer between graphs, even if the graphs are only up to a minor structure difference.

To tackle the challenges above, in this work, we present ResiNet, the first objective-agnostic learning-based method for inductively discovering resilient network topologies. Frist, by formulating the cumulative gain of the objective into RL, ResiNet is agnostic to arbitrary objective function, which makes resilience optimization practical in real applications by incorporating the utility into the resilience. The nature of delayed gratification in RL also allows the agent to bypass local optimums caused by step-wise optimal actions. Second, the fact that graphs have different sizes and arbitrary node permutations makes it challenging to solve transductivity when selecting edges with deep learning. In this scenario, nodes can no longer be represented by their ID since it is computationally infeasible to train a model to handle all isomorphic graphs. Using an attention-based mechanism (Vinyals et al., 2015) to identify edges and an auto-regressive space (Trivedi et al., 2020) to formulate edge selection, an agent trained by ResiNet works on a wide range of graphs with different sizes and permutations.

Deep learning and graph neural networks (GNNs) have been successfully applied to many combinatorial optimization problems on graphs (Khalil et al., 2017; Li et al., 2018a; Karalias & Loukas, 2020). However, as we empirically observed, the combination of GNNs and RL cannot perform well on the task of improving network resilience via edge rewiring. We suspect that this is caused by a dilemma for the expressive power of GNN: If the representation power of the GNN is too weak (e.g., the regular GNN), then it could not distinguish graphs with minor differences and will choose similar actions in consecutive steps of RL. This causes the process to alternate between two graphs, forming an infinite loop. If the representation power is too strong (e.g., SMP, IDGNN), it tends to give similar representations to connected vertices, known as the *oversmoothing phenomenon* (Vignac et al., 2020; You et al., 2021). It is then difficult for the agent to distinguish vertices in a connected component, causing large randomness in the output of the policy. This dilemma may lead to the bad performance of selecting edges using regular GNNs and RL. Therefore, to successfully select two edges from a degree-preserving evolving dynamic graph at each step, the implementation of ResiNet is armed with Filtration enhanced GNN (FireGNN), our technical innovation, to

solve the above dilemma of the expressive power of GNN. FireGNN creates a series of subgraphs (the filtration) by successively removing nodes from the graph, and aggregates the node representations from the subgraphs. It makes the GNN powerful when representing graphs while avoiding oversmoothing as the node information are adequately acquired from a multi-step filtration. This technical innovation is inspired by persistent homology and the approximation of the persistence diagram (Edelsbrunner & Harer, 2008; Aktas et al., 2019; Hofer et al., 2020).

The main contributions of this paper are summarized as follows:

1) We propose ResiNet, a data-driven framework to boost network resilience in a degree-preserving way with moderate loss of the network utility by forming resilience optimization into a objective-agnostic sequential decision process of edge rewiring. Extensive experiments show that with a small number of rewiring operations, ResiNet achieves a near-optimal resilience gain on multiple graphs while balancing network utilities. Existing approaches are outperformed by a large margin.

2) FireGNN, our technical innovation, balances the expressive power and the oversmoothing issue through the graph filtration augmentation. FireGNN could distinguish very similar graphs and distinguish connected vertices at the same time, which is essential to make the combination of GNNs and RL work on the network resilience optimization.

3) ResiNet is the first to improve network resilience in an inductive way, with the specialized auto-regressive permutation-invariant size-variable policy network. Once an agent is trained, it works on a wide range of graphs. The policy network is general enough to be extended to other problems in constrained graph generation.

## 2 RELATED WORKS

**Network attacks and defenses** The problem of attacking a network is characterized by the *target resilience measurement* and the *manipulation type*. Depending on the application, network resilience is defined as a corresponding measure to quantify the network functionality, such as graph connectivity (Grassia et al., 2021; Fan et al., 2020). For particular networks (e.g., scale-free networks), attacks appear in destroying critical nodes and critical connections (Grassia et al., 2021; Fan et al., 2020; Zhao et al., 2021; Zhang et al., 2017; Medya et al., 2020). Heuristic and learning-based attack methods (Holme et al., 2002; Iyer et al., 2013; Grassia et al., 2021; Fan et al., 2020) have been proposed to target the critical subset of the network. To defend against potential attacks, various defense strategies have been proposed to protect the network functionality from crashing and preserve some of its topologies. The commonly used defense manipulations include adding additional edges (Li et al., 2019; Carchiolo et al., 2019), protecting vulnerable edges (Wang et al., 2014) and rewiring two edges (Schneider et al., 2011; Chan & Akoglu, 2016; Buesser et al., 2011). Among these manipulations, edge rewiring fits well to real-world applications as it induces less functionality changes (e.g. degree preserving) to the original network and does not impose additional loads to the vertices (Schneider et al., 2011; Rong & Liu, 2018; Yazıcıoğlu et al., 2015; Zhou & Liu, 2014).

**GNNs for combinatorial optimization problems** The idea of applying GNNs to solve combinatorial optimization problems on graphs is becoming a promising yet challenging topic. Abundant algorithms have been developed and can be classified into three categories. *Supervised learning approach* combines GNNs with heuristics search procedures or solvers to solve classical combinatorial problems, such as graph matching (Bai et al., 2018), graph coloring (Lemos et al., 2019), TSP (Li et al., 2018b; Joshi et al., 2019) and SAT (Wang et al., 2019). These methods require labeled instances and thus are difficult to generalize to large-scale instances directly. *Unsupervised learning approach* (Karalias & Loukas, 2020) trains GNNs to parametrize a probability distribution over sets. The probabilistic proof achieves the existence of feasible solutions. After that, the derandomized method is applied to decode the desired solutions. However, meeting complex constraints have not been supported yet. *Reinforcement learning approach* formulates a problem as a Markov decision process with a proper reward function to guide the search towards an optimal solution. Some classical CO problems have achieved remarkable results like TSP (Fu et al., 2020; Khalil et al., 2017), Vehicle Routing Problem (Nazari et al., 2018; Peng et al., 2020; Yu et al., 2019), SAT (Yolcu & Póczos, 2019) and max-cut (Khalil et al., 2017). In this paper, we adopt the RL based approach but with a specialized policy network.

**Positioning of this work**    In this paper, we use only the edge rewiring operation to defend against degree-based and centrality-based attacks while preserving the network utility.

**Extended related works**    The related works on *Network resilience and utility*, *multi-views graph augmentation for GNNs*, and *deep graph generation models* are deferred to Appendix A.

## 3    PROPOSED APPROACH: RESINET

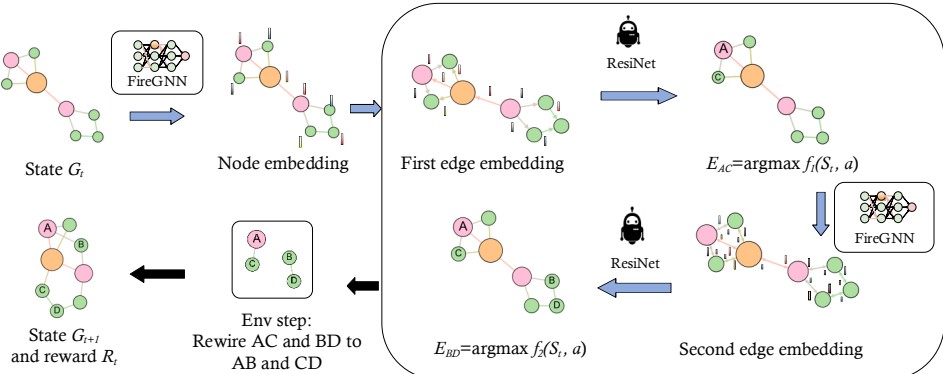

Figure 2: Overview of the policy architecture of ResiNet to select two rewiring edges.

In this section, we first present the problem formulation of maximizing network resilience while preserving utility performance. We then discuss the graph resilience-aware environment design and give the graph policy network which guides the process of edge rewiring. The problem of boosting network resilience is formulated as a reinforcement learning task by iteratively rewiring the edges.

### 3.1    PROBLEM DEFINITION

An undirected graph is defined as $G = (V, E)$, where $V = \{1, 2, \ldots, N\}$ is the set of $N$ nodes, $E$ is the set of $M$ edges, $A \in \{0, 1\}^{N \times N}$ is the adjacency matrix, and $F \in \mathbb{R}^{N \times d}$ is the $d$-dimensional node feature matrix.[2] The degree of a node is defined as $d_i = \sum_{j=1}^{N} A_{ij}$, and a node with degree 0 is called an isolated node.

Let $\mathbb{G}_G$ denote the set of graphs with the same node degrees as $G$. Given the network resilience function $\mathcal{R}(G)$ and the utility function $\mathcal{U}(G)$, the objective of boosting the resilience of $G$ is to find a target graph $G^\star \in \mathbb{G}_G$, which maximizes the network resilience while preserving the network utility. Formally, the problem of maximizing network resilience is formulated as

$$G^\star = \arg\max_{G' \in \mathbb{G}_G} \alpha \cdot \mathcal{R}(G') + (1 - \alpha) \cdot \mathcal{U}(G'),$$

where $\alpha \in \mathbb{R}$ is the scalar weight that balances the resilience and the utility.

Two examples of resilience functions, including the graph connectivity-based measurement (Schneider et al., 2011) and the spectrum-based measurement (e.g., adjacency matrix spectrum and Laplacian matrix spectrum), and two examples of utility functions, including global efficiency and local efficiency, are given in Appendix B. Our formulation could generalize to other definitions of resilience and utility.

### 3.2    ROBUST GRAPH GENERATION VIA EDGE REWIRING AS MARKOV DECISION PROCESS

To satisfy the constraint of preserving the node degree, the resilience optimization of a given graph is based on edge rewiring. We formulate this process into the MDP framework. The Markov property

---

[2]Without loss of generality, unweighted graphs are considered. Our framework can be extended to weighted graphs with the weight as an additional edge feature.

denotes the fact that the graph obtained at time step $t+1$ relies only on the graph at time step $t$ and the rewire operation. As is illustrated in Figure 2, the environment performs the resilience optimization in an auto-regressive step-wise way through a sequence of edge rewiring actions guided by ResiNet. Given an input graph, the agent first decides whether to terminate if or not. If it chooses not to terminate, it selects one edge from the graph to remove, receives the very edge it just selected as the regression signal, and then selects another edge to remove. Four nodes of these two removed edges are re-combined, forming two new edges to be added to the graph. The optimization process repeats until the agent decides to terminate. The details of the design of the state, the action, the transition dynamics, and the reward are presented as follows.

**State** The fully observable state of the environment is formulated as $S_t = G_t$, where $G_t$ is the current input graph at time step $t$. The node feature initialization strategy is given in Appendix C.3.

**Action** ResiNet is equipped with a node permutation-invariant, variable-dimensional action space. Given a graph $G_t$, the action $a_t$ is to select two edges and the rewiring order. The agent first chooses an edge $e_1 = AC$ and a direction $A \rightarrow C$. Then conditioning on the state, $e_1$, and the direction the agents chooses an edge $e_2 = BD$ such that $AB, CD, AD, BC \notin E$ and a direction $B \rightarrow D$. The heads of the two edges reconnect as a new edge $AB$ and so does the tail $CD$. As $A \rightarrow C$, $B \rightarrow D$ and $C \rightarrow A$, $D \rightarrow B$ refer to the same rewiring operation, the choice of the direction of $e_1$ is randomized (this randomized bit is still an input of choosing $e_2$), as is shown in Figure 3. This effectively reduces the size of the action space by half. In this way, The action space is the set of all feasible pairs of $(e_1, e_2) \in E^2$, which has a variable size no larger than $2|E|(|E| - 1)$.

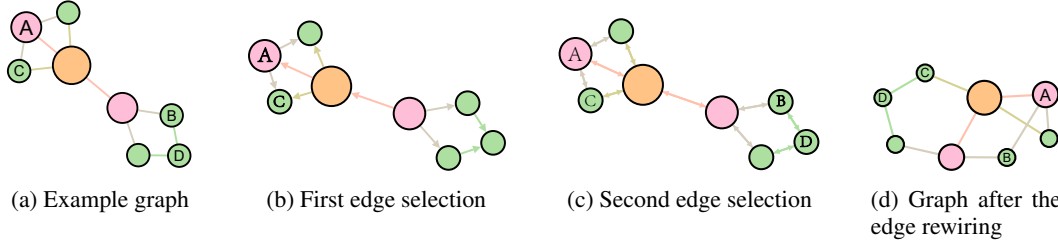

(a) Example graph      (b) First edge selection      (c) Second edge selection      (d) Graph after the edge rewiring

Figure 3: Illustration of the edge rewiring and the rewiring order. (a) The undirected example graph as the state; (b) Edges with random directions are candidates when selecting the first edge; (c) All $2M$ directed edges can be candidates when selecting the second edge, subject to that it forms a rewiring with the first edge; (d) The resultant graph after the rewiring, with the removal of $AC$, $BD$ and the introduction of $AB$, $CD$.

**Transition dynamics** By the formulation of the action space, if the agent does not terminate at step $t$, the selected action must form an edge rewiring. This edge rewiring is executed by the environment and the graph transits to the new graph.

Note that in some other works, infeasible operations are also included in the action space (to make the action space constant through the process) (You et al., 2018; Trivedi et al., 2020). In these works, if the operation is infeasible, it is not executed and the state is not changed. This reduces the sample efficiency, causes biased gradient estimation (Huang & Ontañón, 2020), and makes the process to be prone to stuck at the state (which requires manually disabling the specific action in the next step). ResiNet takes the advantage of the action space composed of only feasible operations.

**Reward** ResiNet desires to optimize the resilience while preserving the utility, forming a complicated and possibly unknown objective function. Despite this, by Wakuta (1995), an MDP that maximizes a complicated objective is up to an MDP that maximizes the linear combination of resilience and utility for some coefficient factor. This fact motivates us to design the reward as the step-wise gain of such a linear combination as

$$R_t = \alpha \cdot \mathcal{R}(G_{t+1}) + (1 - \alpha) \cdot \mathcal{U}(G_{t+1}) - (\alpha \cdot \mathcal{R}(G_t) + (1 - \alpha) \cdot \mathcal{U}(G_t)),$$

where $\mathcal{R}(G)$ and $\mathcal{U}(G)$ are the resilience and the utility functions, respectively. The cumulative reward $\sum_{t=0}^{T-1} R_t$ up to time $T$ is then the total gain of such a linear combination.

### 3.3 EDGE REWIRING POLICY

Having presented the details of the resilient graph generation environment, in this section, we describes the architecture of ResiNet in details. ResiNet is a policy network trained to maximize the cumulative reward of the agent in the graph resilience-aware environment. At time step $t$, the policy network takes the current graph $G_t$ as input and outputs an action $a_t$ which represents the two rewired edges, leading to the new state $G_{t+1}$ and the reward $R_t$.

#### 3.3.1 FIREGNN VIA GRAPH FILTRATION ARGUMENTATION

Motivated by the graph filtration in persistent homology (Edelsbrunner & Harer, 2008), we propose to enhance the expressivity of GNN via the graph filtration process by augmenting an input graph $G$. To avoid the computational cost of calculating the persistence diagram at each step, the similarity between the filtration process and resultant subgraphs after each attack motivates us to construct a sequence of nested subgraphs of $G$ as an approximate filtration, such that

$$(V, \emptyset) = G^{(0)} \subset G^{(1)} \subset \cdots \subset G^{(N)} = G\,,$$

where $G^{(k)}$ denotes the remaining graph after removing $N - k$ nodes in a particular task-dependent order, $G^{(N)}$ is the original graph, and $G^{(0)}$ is an empty graph with no edge.

The sequence of nested subgraphs of $G$ is termed the filtrated graph $G$. Regular GNN only operates on the original graph $G$ to obtain the node embedding for each node $v_i$ as

$$h(v_i) = \phi(G^{(N)} = G)_i\,,$$

where $\phi(\cdot)$ denotes a standard GNN model.

**Node embedding**  In FireGNN, by using the top $K + 1$ subgraphs in a graph filtration, the final node embedding $h(v_i)$ of $v_i$ is obtained by

$$h(v_i) = \mathrm{Agg}_N \left( h^{(N-K)}(v_i), h^{(N-K+1)}(v_i), \ldots, h^{(N-1)}(v_i), h^{(N)}(v_i) \right)\,,$$

where $\mathrm{Agg}_N(\cdot)$ denotes a node-level aggregation function, $h^{(k)}(v_i)$ is the node embedding of $i$ in the $k$-th subgraph $G^{(k)}$, and $K \in [N]$. In practice, $h^{(k)}(v_i)$ is discarded when calculating $h(v_i)$ if $v_i$ is isolated or not included in $G^{(k)}$.

**Edge embedding**  The directed edge embedding $h^{(k)}(e_{ij})$ of the edge from node $i$ to node $j$ in each subgraph is obtained by combining the embeddings of the two end vertices in $G^{(k)}$ as

$$h^{(k)}(e_{ij}) = m_f \left( \mathrm{Agg}_{N \to E} \left( h^{(k)}(v_i), h^{(k)}(v_j) \right) \right)\,,$$

where $\mathrm{Agg}_{N \to E}(\cdot)$ denotes an aggregation function for obtaining edge embedding from two end vertices (typically chosen from `min`, `max`, `sum`, `difference`, and `multiplication`). $m_f(\cdot)$ is a multilayer perceptron (MLP) model that ensures the consistent dimension of edge embedding and graph embedding.

The final embedding of the directed edge $e_{ij}$ of the filtrated graph $G$ is given by

$$h(e_{ij}) = \mathrm{Agg}_E \left( h^{(N-K)}(e_{ij}), h^{(N-K+1)}(e_{ij}), \ldots, h^{(N-1)}(e_{ij}), h^{(N)}(e_{ij}) \right)\,,$$

where $\mathrm{Agg}_E(\cdot)$ denotes an edge-level aggregation function.

**Graph embedding**  With the node embedding $h^{(k)}(v_i)$ of each subgraph available, the graph embedding $h^{(k)}(G)$ of each subgraph is calculated by a readout functions (e.g., `mean`, `sum`) on all non-isolated nodes in $G^{(k)}$ as

$$h^{(k)}(G) = \mathrm{READOUT} \left( h^{(k)}(v_i) \right) \ \forall v_i \in G^{(k)} \text{ and } d_i^{(k)} \geq 0\,.$$

The final graph embedding of the filtrated graph $G$ is given by

$$h(G) = \mathrm{Agg}_G \left( h^{(N-K)}(G), h^{(N-K+1)}(G), \ldots, h^{(N-1)}(G), h^{(N)}(G) \right)\,,$$

where $\text{Agg}_G(\cdot)$ denotes a graph-level aggregation function.

By observing a sequence of nested subgraphs of $G$, a filtration process transforms an input graph from the static graph to a sequences of subgraphs, which grants GNNs the capability to observe how $G$ evolves and improve the expressivity in node, edge, and graph levels. We term the **Fil**trated graph **e**nhanced **GNN** as FireGNN.

### 3.3.2 POLICY NETWORK

After obtaining the directed edge embedding $h(e_{ij}) \in \mathbb{R}^{2|E| \times d}$ and the graph embedding $h(G) \in \mathbb{R}^d$ from the proposed FireGNN, armed with our specially designed autoregressive node permutation-invariant dimension-variable action space, we can select the edge rewiring operation for graphs with arbitrary sizes and node-permutations. The detailed architecture of ResiNet and the mechanism of obtaining the action $a_t$ are presented as follows.

**Auto-regressive latent action selection** An edge rewiring action $a_t$ at time step $t$ involves the prediction of the termination probability $a_t^{(0)}$ and the selection of two edges ($a_t^{(1)}$ and $a_t^{(2)}$) and the rewiring order. The action space of $a_t^{(0)}$ is binary, however, the selection of two edges imposes a huge action space in $O(E^2)$, which is too expensive to sample from even for a small graph. Instead of selecting two edges simultaneously, we decompose the joint action $a_t$ into $a_t = (a_t^{(0)}, a_t^{(1)}, a_t^{(2)})$, where $a_t^{(1)}$ and $a_t^{(2)}$ are two existing edges which do not share any common node (recall that $a_t^{(1)}$ and $a_t^{(2)}$ are directed edges for an undirected graph). Thus the probability of $a_t$ is formulated as

$$\mathbb{P}(a_t|s_t) = \mathbb{P}(a_t^{(0)}|s_t)\mathbb{P}(a_t^{(1)}|s_t, a_t^{(0)})\mathbb{P}(a_t^{(2)}|s_t, a_t^{(0)}, a_t^{(1)}) \,.$$

**Predicting the termination probability** The first policy network $\pi_0(\cdot)$ takes the graph embedding as input and outputs the probability distribution of the first action that decides to terminate or not as

$$\mathbb{P}(a_t^{(0)}|s_t) = \pi_0(h(G)) \,,$$

where $\pi_0(\cdot)$ is implemented by a two layer MLP. Then $a_t^{(0)} \sim \text{Ber}(\mathbb{P}(a_t^{(0)}|s_t)) \in \{0, 1\}$.

**Selecting two edges** If the signal $a_t^{(0)}$ given by the agent decides to continue to rewire, two edges are then selected in an auto-regressive way. The signal of continuing to rewire $a_t^{(0)}$ is input to the selection of two edges as a one-hot encoding vector $l_c$. The second policy network $\pi_1(\cdot)$ takes the graph embedding and $l_c$ as input and outputs a latent vector $l_1 \in \mathbb{R}^d$. The pointer network (Vinyals et al., 2015) is used to measure the proximity between $l_1$ and each edge embedding $h(e_{ij})$ in $G$ to obtain the first edge selection probability distribution. Then, to select the second edge, the graph embedding $h(G)$ and the first selected edge embedding $h(e_t^{(1)})$ and $l_c$ are concatenated and fed into the third policy network $\pi_2(\cdot)$. $\pi_2(\cdot)$ obtains the latent vector $l_2$ for selecting the second edge using a respective pointer network. The process can be formulated as

$$l_1 = \pi_1([h(G), l_c])$$
$$\mathbb{P}(a_t^{(1)}|s_t, a_t^{(0)}) = f_1(l_1, h(e_{ij}))), \forall e_{ij} \in E$$
$$l_2 = \pi_2([h(G), h_{e_t^{(1)}}, l_c])$$
$$\mathbb{P}(a_t^{(2)}|s_t, a_t^{(1)}, a_t^{(0)}) = f_2(l_2, h(e_{ij}))), \forall e_{ij} \in E \,,$$

where $\pi_i(\cdot)$ is implemented by a two layer MLP model. $[\cdot, \cdot]$ denotes the concatenation operator, $h_{e_t^{(1)}}$ is the embedding of the first selected edge at time step $t$, and $f_i(\cdot)$ is a pointer network.

## 4 EXPERIMENTS

In this section, we demonstrate the advantage of ResiNet in optimizing network resilience, generalizing to unseen graphs, and accommodating multiple resilience and utility metrics. The experimental results show that the optimization of ResiNet is achieved with fewer edge rewiring operations.

## 4.1 EXPERIMENTAL SETTINGS

**Datasets**  Synthetic datasets and the real EU power network (Zhou & Bialek, 2005) are used to demonstrate the performance of ResiNet in both transductive and inductive settings. The details of data generation and the statistics of the datasets are presented in Appendix C.1.

**Baselines**  We compare ResiNet with existing graph resilience optimization algorithms, including the hill climbing (HC) (Schneider et al., 2011), the greedy algorithm (Chan & Akoglu, 2016), the simulated annealing (SA) (Buesser et al., 2011), and the evolutionary algorithm (EA) (Zhou & Liu, 2014). It should be noted that all traditional baselines are transductive. Since there is no neural-version baseline, we provide one by replacing FireGNN with a regular GNN. The ResiNet's training setup is detailed in Appendix C.2. All algorithms are repeated for three random seeds using default hyper-parameters to obtain the averaged performance.

## 4.2 COMPARISONS TO THE BASELINES

In this set of experiments, we first compare ResiNet to the baselines in optimizing the combination of resilience and utility with weight coefficient $\alpha \in \{0, 0.5\}$. The graph connectivity-based metric is used as the resilience metric and the global efficiency is used as the utility metric. The test is conducted on 4 transductive settings and 3 inductive settings.

Table 1 records the metric gain and the required number of rewiring of different methods under the same maximal rewiring budget. ResiNet outperforms all baselines consistently on all datasets. Note that this performance may be achieved under a much fewer number of rewiring operations, such as the BA-15 with $\alpha = 0$. In contrast, even approximately searching for all possible new edges, the greedy algorithm is still trapped into a local optimum by maximizing the one-step resilience gain. For SA, the initial temperature and the temperature decay rate need to be carefully tuned for each network. EA performs worse with a limited rewiring budget due to the numerous rewiring operations required in the internal process (e.g., the crossover operator). It should be noted that the poor performance of a combination of regular GNNs and RL (GNN+RL), compared to ResiNet, validates the challenges of improving network resilience via edge rewiring and indicates the effectiveness of ResiNet with FireGNN.

In general, the metric gain is more significant for larger graphs, which implies that it is reasonable and cost-effective to improve the resilience of large networks. All baselines are compared under the same rewiring budget of 20 since each edge rewiring introduces economic costs for real systems. It should be noted that solutions under a large rewiring budget like 200 will not be applicable in practice due to the tremendous rewiring cost. We record the performance and speed of each algorithm for a maximal rewiring budget of 200 in Appendix Table 4.

## 4.3 GENERALIZATION

In this section, we conduct extensive experiments to show that ResiNet generalizes to different utility and resilience metrics, various attack strategies, and unseen graphs. We do not compare ResiNet with other algorithms on the inductive setting because existing algorithms are transductive.

To explore the complicated objective of resilience and utility, the BA-15 network is taken as an example to be optimized by ResiNet to obtain the approximate Pareto frontier. The Pareto points are shown in Figure 8 to denote the optimum under different objectives. Surprisingly, the initial gain of resilience (from around 0.21 to around 0.24) is without loss of the utility, which incentivizes almost every network to conduct such optimization at least to some extent when feasible.

To demonstrate that ResiNet can learn from the networks for accommodating different utility and resilience metrics, we conduct experiments based on the BA-15 and EU datasets by using multiple different resilience metrics and different utility metrics (details are in Appendix B). The results are included in Appendix D.2. The network structures are visualized in Figure 7 and Figure 10.

To demonstrate the inductivity of ResiNet, we first train ResiNet on two different datasets, and the data setting is recorded in Table 2. We evaluate its performance on an individual test dataset. These datasets are unknown to ResiNet and have not been observed during the training process. Fine-tuning is not allowed. We report the averaged resilience gain for the graphs with the same number

Table 1: Resilience optimization algorithm under the fixed maximal rewiring number budget of 20. We report the weighted combination of the graph connectivity-based resilience and the network efficiency improvement (in percentage) with the weighted coefficient $\alpha \in \{0, 0.5\}$. Results are averaged over 3 runs. We report the rewiring number inside the bracket.

| Method | $\alpha$ | BA-15 | BA-50 | BA-100 | EU |
|---|---|---|---|---|---|
| HC | 0 | 26.8 (10) | 30.0 (20) | 24.1 (20) | 19.8 (20) |
| | 0.5 | 18.6 (11.3) | 22.1 (20) | 14.9 (20) | 16.3 (20) |
| SA | 0 | 21.6 (17.3) | 11.9 (20) | 12.5 (20) | 14.9 (20) |
| | 0.5 | 16.8 (19.0) | 11.4 (20) | 13.4 (20) | 14.0 (20) |
| Greedy | 0 | 23.5 (6) | 48.6 (13) | 64.3 (20) | 0.5 (3) |
| | 0.5 | 5.3 (15) | 34.7 (13) | 42.7 (20) | 0.3 (3) |
| EA | 0 | 8.5 (20) | 6.4 (20) | 4.0 (20) | 8.2 (20) |
| | 0.5 | 6.4 (20) | 4.7 (20) | 2.8 (20) | 9.3 (20) |
| GNN+RL | 0 | 13.7 (2) | 0 (1) | 0 (1) | 9.0 (20) |
| | 0.5 | 10.9 (2) | 0 (1) | 0 (1) | 2.1 (20) |
| ResiNet | 0 | **35.3** (6) | **61.5** (20) | **70.0** (20) | **54.2** (20) |
| | 0.5 | **26.9** (20) | **53.9** (20) | **53.1** (20) | **51.8** (20) |

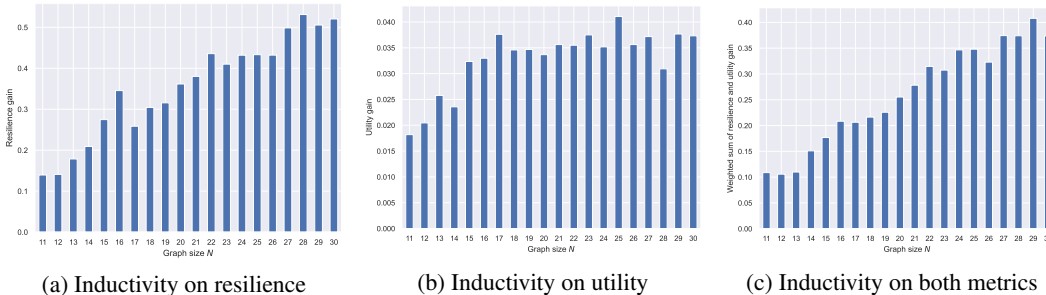

(a) Inductivity on resilience     (b) Inductivity on utility     (c) Inductivity on both metrics

Figure 4: The inductive ability of ResiNet on the test dataset (BA-10-30) when optimizing (a) network resilience, (b) network utility, and (c) their combination.

of nodes for each dataset. The result of BA-10-30 is shown in Figure 4 and the results of the other datasets are deferred to Appendix D.2. Figure 4 shows a nearly linear improvement of resilience with the increasing of graph size, which is also consistent with the results in the transductive setting that larger graphs usually have a larger room to improve.

## 5   CONCLUSION AND FUTURE WORKS

In this work, we propose a general, learning-based framework, ResiNet, for the inductive discovery of resilient network topologies. The framework is practically feasible as it also preserves the utility of the networks. ResiNet is the first to formulate the problem of boosting network resilience as an MDP, with a specialized auto-regressive permutation-invariant variable action space. This variant of MDP allows the agent to handle graphs of different sizes and isomorphisms. Once an agent is trained, it inductively works on a wide range of graphs and is ready to be deployed to real systems. Our technical innovation, FireGNN, is a powerful technique for node and graph representations, which is a suitable replacement to other GNNs, especially in RL. It balances the expressive power and the oversmoothing issue of GNNs and could distinguish graphs and connected nodes at the same time. Both ResiNet and FireGNN are general enough to be applied to a variety of problems in graph theory, network theory, combinatorial optimization, machine learning, and their applications.

An interesting direction of future work is to obtain a better approximation algorithm for the persistence diagram. The current implementation includes an average operation across the entire filtration, which is computationally expensive when the graph is large and forces us to truncate the filtration in our implementations. An efficient and accurate approximation algorithm will grant FireGNN better feasibility in real applications.

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

## A    EXTENDED RELATED WORK

### Network resilience and utility

Network utility refers to the system's quality to provide a certain service, for example transmitting electricity in power networks and transmitting packages in routing networks. A popular metric for network utility is the network efficiency (Latora & Marchiori, 2003; Boccaletti et al., 2006). Network resilience measures the ability of preventing the utility loss under failures and attacks. Our goal is to improve both network resilience and utility by network structure manipulations.

In many previous works, despite that the resilience could be improved, the utility may dramatically drop at the same time (Li et al., 2019; Carchiolo et al., 2019; Wang et al., 2014; Schneider et al., 2011; Chan & Akoglu, 2016; Buesser et al., 2011). This is contradicting with the idea behind improving network resilience and will be infeasible in real-world applications.

### Multi-views graph augmentation for GNNs

Multi-views graph augmentation is one efficient way to improve expressive power or combine domain knowledge, which is adapted based on the task's prior (Hu et al., 2020). For example, GCC (Qiu et al., 2020) generates multiple subgraphs from the same ego network. DGI (Velickovic et al., 2019) maximizes the mutual information between global and local information. GCA (You et al., 2020) adaptively incorporates various priors for topological and semantic aspects of the graph. (Hassani & Khasahmadi, 2020) contrasts representations from first-order neighbors and a graph diffusion. In our problem, two desired properties are different edge representations and the capability to predict accurate network resilience gain. Motivated by the calculation process of persistent homology (Edelsbrunner & Harer, 2008), we apply the filtration process to enhance the expressive power of GNNs.

### Deep graph generation models

Deep graph generation models learn the distribution of given graphs and generate more novel graphs, which could be classified as unconditional generation methods and conditional generation methods

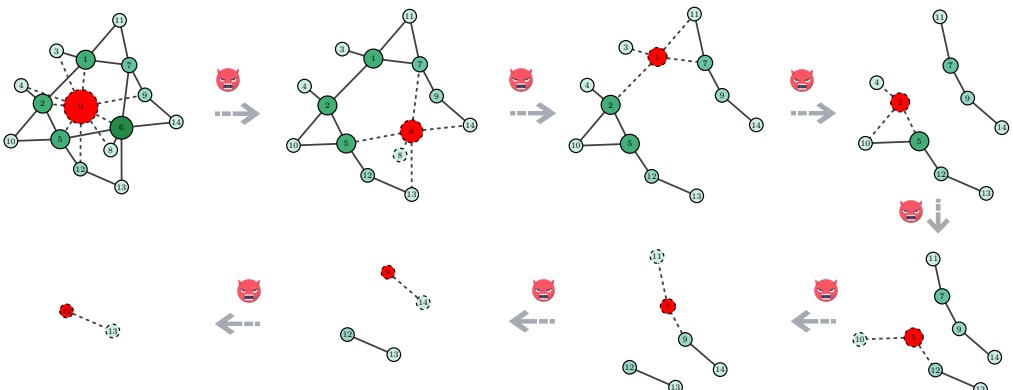

Figure 5: Illustration of the graph connectivity-based measurement calculation with a scale-free example graph under node degree-based malicious attack. Each snapshot demonstrates the attack on the red node, which removes the red node and its adjacent (dashed) edges, leading to a next remnant graph. The attacker recalculates the node degree and repeatedly seeks for the remaining node with a maximum degree as the next target. In the end, the entire graph becomes disconnected when each node cannot reach other nodes.

(Guo & Zhao, 2020). Specifically, edge transformation conditioned on another graph (a.k.a. graph translation) is more relevant to our work since the edge rewiring process can be regarded as operations on the previous graph via edge addition and deletion. Some works utilize the encoder-decoder framework by learning latent representation of the input graph through the encoder and then generating the target graph through the decoder. For example, GCPN (You et al., 2018) incorporates chemistry domain rules on molecular graph generation. GT-GAN (Guo et al., 2018) proposes a GAN-based model on malware cyber-network synthesis. GraphOpt (Trivedi et al., 2020) learns an implicit model to discover an underlying optimization mechanism of the graph generation using inverse reinforcement learning. Although exploiting a similar encoder-decoder framework, our approach differs from existing works by the rigid constraint of the studied task: node degree preservation, which imposes additional requirements for the expressive power of GNNs.

## B  DEFINITIONS OF DIFFERENT OBJECTIVE FUNCTIONS

In this section, we present resilience definitions and utility definitions used in our experiments.

### B.1  RESILIENCE DEFINITIONS

Three kinds of resilience metrics are considered:

- The graph connectivity-based measurement (Schneider et al., 2011) is defined as

$$\mathcal{R}(G) = \frac{1}{N} \sum_{q=1}^{N} s(q),$$

  where $s(q)$ is the fraction of nodes in the largest connected remaining graph after removing $q$ nodes from graph $G$ according to certain attack strategy, as shown in Figure 5. The range of possible values of $\mathcal{R}$ is $[1/N, 1/2]$, where these two extreme values correspond to a star network and a fully connected network, respectively.

- The spectral radius ($\mathcal{SR}$) denotes the largest eigenvalue $\lambda_1$ of an adjacency matrix.

- The algebraic connectivity ($\mathcal{AC}$) represents the second smallest eigenvalue of the Laplacian matrix of $G$.

## B.2 Utility definitions

In this paper, the global and local communication efficiency are used as two measurements of the network utility, which are widely applied across diverse applications of network science, such as transportation and communication networks (Latora & Marchiori, 2003; Boccaletti et al., 2006).

The average efficiency of a network $G$ is defined as inversely proportional to the average over pairwise distances (Latora & Marchiori, 2001) as

$$E(G) = \frac{1}{N(N-1)} \sum_{i \neq j \in V} \frac{1}{d(i,j)},$$

where $N$ denotes the total nodes in a network and $d(i,j)$ denotes the length of the shortest path between a node $i$ and another node $j$.

Given the average efficiency, we can calculate the global efficiency and local efficiency.

- The global efficiency of a network $G$ is defined as (Latora & Marchiori, 2001; 2003)

$$E_{global}(G) = \frac{E(G)}{E(G^{ideal})},$$

where $G^{ideal}$ is the "ideal" fully-connected graph on $N$ nodes, and the range of $E_{global}(G)$ is [0, 1].

- The local efficiency of a network $G$ measures a local average of pairwise communication efficiencies and is defined as (Latora & Marchiori, 2001)

$$E_{local}(G) = \frac{1}{N} \sum_{i \in V} E(G_i),$$

where $G_i$ is the local subgraph including only of a node $i$'s one-hop neighbors, but not the node $i$ itself. The range of $E_{local}(G)$ is [0, 1].

## C Implementation details of ResiNet

In this section, we provide the implementation details of ResiNet, including dataset, network structure training strategies and node feature construction. Moreover, we discussed the role of FireGNN in ResiNet and analyze current limitations of ResiNet and future work.

### C.1 Dataset

We first present the data generation strategies. Synthetic datasets are generated using the Barabasi-Albert (BA) model (known as scale-free graphs) (Albert & Barabási, 2002), with the graph size varying from $|N|$=10 to $|N|$=200. During the data generation process, each node is connected to two existing nodes. BA graphs are chosen since they are vulnerable to malicious attacks and are commonly used to test network resilience optimization algorithms(Bollobás & Riordan, 2004). We test the performance of ResiNet on both transductive and inductive settings.

- **Transductive setting**    The algorithm is trained and tested on the same network.
  - Randomly generated synthetic BA networks, denoted by BA-$m$, are adopted to test the performance of ResiNet on networks of various sizes, where $m \in \{15, 50, 100\}$ is the graph size.
  - The real EU power network (Zhou & Bialek, 2005) is used to validate the performance of ResiNet on real networks.
- **Inductive setting**    Two groups of synthetic BA networks denoted by BA-$m$-$n$ are randomly generated to test ResiNet's inductivity, where $m$ is the minimal graph size, and $n$ indicates the maximal graph size. We first randomly generate the fixed number of BA networks as the training data to train ResiNet and then evaluate ResiNet's performance directly on the test dataset without any additional optimization. We only consider ResiNet for the inductive setting since other methods are all transductive.

Table 2 summarizes the statistics of each dataset.

Table 2: Statistics of graphs used for resilience maximization. Both transductive and inductive settings ($\star$) are included. Consistent with our implementation, we report the number of edges by transforming undirected graphs to directed graph so that the edge rewiring has the fixed execution order. For the inductive setting, we report the maximum number of edges. The action space size of the edge rewiring is measured by $2|E|^2$.

| Dataset | Node | Edge | Action Space Size | Train/Test | Setting |
|---|---|---|---|---|---|
| BA-15 | 15 | 54 | 5832 | Null | Transductive |
| BA-50 | 50 | 192 | 73728 | Null | Transductive |
| BA-100 | 100 | 392 | 307328 | Null | Transductive |
| EU | 217 | 640 | 819200 | Null | Transductive |
| BA-10-30 ($\star$) | 10-30 | 112 | 25088 | 1000/500 | Inductive |
| BA-20-200 ($\star$) | 20-200 | 792 | 1254528 | 4500/360 | Inductive |

## C.2 RESINET SETUP

In this section, we provide detailed parameter setting and training strategies for ResiNet.

Our proposed FireGNN is used as the graph encoder in ResiNet, including a 5-layer defined GIN (Xu et al., 2019) as the backbone. The hidden dimensions for node embedding and graph embedding in each hidden layer are set to 64 and the SeLU activation function (Klambauer et al., 2017) is used after each message passing propagate (Li et al., 2020a). Graph normalization strategy is adopted to stabilize the training of GNN (Cai et al., 2020). The jumping knowledge network (Xu et al., 2018) is used to aggregate node features from different layers of the GNN.

The overall policy is trained by using the highly tuned implementation of proximal policy optimization (PPO) algorithm (Schulman et al., 2017). Several critical strategies for stabilizing and accelerating the training of ResiNet are used, including advantage normalization (Andrychowicz et al., 2021), the dual-clip PPO (the dual clip parameter is set to 10) (Ye et al., 2020), and the usage of different optimizers for policy network and value network. Additionally, since the step-wise reward range is small (around 0.01), we scale the reward by a factor of 10 to aim the training of ResiNet. The policy head model and value function model use two separated FireGNN encoder networks with the same architecture. ResiNet is trained using two separate Adam optimizers (Kingma & Ba, 2015) with batch size 256 and a linearly decayed learning rate of 0.0007 for the policy network and a linearly decayed learning rate of 0.0007 for the value network. The aggregation function of FireGNN is defined as an attention mechanism-based linear weighted combination.

## C.3 NODE FEATURE CONSTRUCTION

The widely-used node degree feature cannot significantly benefit the network resilience optimization of a single graph due to the degree-preserving rewiring. Therefore, we construct node features for each input graph to aid the transductive learning and inductive learning, including

- The 8-dimensional position embedding originating from the Transformer (Vaswani et al., 2017) is adopted as the measurement of the vulnerability of each node under attack. If the attack order is available, we can encode it into the position embedding. If the attack order is unknown, node degree, node betweenness, and other node priority metrics can be used for approximating the node importance in practice.
- The distance encoding strategy (Li et al., 2020b).
- One-hot encoding denoting whether the attack causes the node removal or just because all adjacent nodes are removed.

## C.4 BASELINE SETUP

All baselines share the same action space with ResiNet, and they uses the same action masking strategy to block invalid actions as ResiNet does. The maximal objective evaluation is consistent for all algorithms. Other settings of baselines are consistent with the default values in their paper.

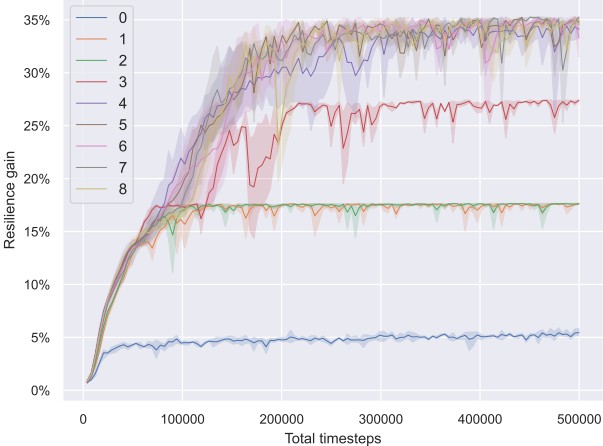

Figure 6: Experiments on the BA-15 dataset to study the effect of the filtration order on FireGNN in maximizing resilience. Results are averaged over 3 runs.

The early-stopping strategy is used for baselines, which means the search process terminates if no improvement is obtained in successive 1000 objective function calling trials.

## D    ABLATION STUDY AND EXTENDED EXPERIMENTAL RESULTS

In this section, we provide the ablation study about the importance of FireGNN and present additional experimental results.

### D.1    FireGNN

In FireGNN, the total number of subgraphs involved in calculating the final node embedding, edge embedding, and graph embedding, are determined by the filtration order $K \in [0, N-1]$, where $K = 0$ means only the original graph is used and $K = N - 1$ means there are total $N$ subgraphs used. We run a grid search to explore the effect of the filtration order on ResiNet's performance. As shown in Figure 6, ResiNet only achieves a minor gain of around 5% without FireGNN. The performance improved significantly with $K$ greater than zero, and ResiNet obtained the optimal resilience gain of about 35 % on BA-15 when $K \geq 5$. We only report the performance when $K \leq 8$ since the BA-15 loses all connections when more than eight critical nodes are removed.

Besides the BA-15 dataset, we also conducted the ablation experiments of FireGNN on large datasets, including the BA-50, BA-100, and EU power network. We found that without FireGNN, ResiNet hardly learns to find a positive gain, so it often chooses to stop at the first step, which may further validate the effectiveness of FireGNN.

It is worth pointing out that the task-dependent order of node removal will affect the filtration. For example, the scale-free networks lost all connections under the degree-based attack on half nodes on average, while it maintains its most topologies under some other attacks. For the network resilience task, we are inspired by the definition of the graph connectivity-based resilience metric, with the adaptive node degree centrality to determine the node removal order. Nodes with larger degrees are first removed, and the ties are broken by arbitrary.

We provide possible reasons why FireGNN is critical for the network resilience task as

- Each rewire only changes the graph by four edges, so the graph embedding and the node embedding may not change significantly at two steps to provide essential information for the RL agent to make new accurate decisions. For example, we empirically found that the RL agent can stick into an action loop, which means that after the rewiring of $A$-$C$ and $B$-$D$ to obtain $A$-$B$ and $C$-$D$ at step $t$, the agent may choose to rewire the new forming edges back at step $t + 1$, forming an infinite loop.

Table 3: Performance gain (in percentage) of ResiNet in optimizing varying objectives on the BA-15 network. All objectives are optimized with the same hyper-parameters, which means that we did not tune hyper-parameters for objectives except for $R_D$.

| Objective | Gain (%) | Objective | Gain(%) |
|---|---|---|---|
| $\mathcal{R}_D$ | 35.3 | $\mathcal{R}_B$ | 14.6 |
| $\mathcal{SR}_D$ | 15.3 | $\mathcal{SR}_B$ | 15.3 |
| $\mathcal{AC}_D$ | 48.2 | $\mathcal{AC}_B$ | 43.2 |
| $\mathcal{R}_D + E_{global}$ | 14.2 | $\mathcal{R}_B + E_{global}$ | 13.1 |
| $\mathcal{SR}_D + E_{global}$ | 14.6 | $\mathcal{SR}_B + E_{global}$ | 15.0 |
| $\mathcal{AC}_D + E_{global}$ | 34.0 | $\mathcal{AC}_B + E_{global}$ | 31.3 |
| $\mathcal{R}_D + E_{local}$ | 24.4 | $\mathcal{R}_B + E_{local}$ | 39.4 |
| $\mathcal{SR}_D + E_{local}$ | 17.3 | $\mathcal{SR}_B + E_{local}$ | 21.2 |
| $\mathcal{AC}_D + E_{local}$ | 9.4 | $\mathcal{AC}_B + E_{local}$ | 15.1 |

- The dominating nodes may potentially be regarded as outliers which can shade other nodes' unique features, which is problematic for the agent's learning to distinguish different nodes. By removing the dominating nodes, the residual nodes in each subgraph have the freedom to express their salient features.

- It might also be potentially valuable for other edge embedding-based tasks, such as graph generation, recommendation, and link prediction.

## D.2 MORE EXPERIMENTAL RESULTS

This section provides additional experimental results, including the optimization with different resilience and utility metrics and validating ResiNet's inductivity on larger datasets. Finally, we analyze the current limitations of ResiNet.

### D.2.1 LEARNING TO BALANCE MORE UTILITY AND RESILIENCE METRICS

As shown in Figure 7, we conduct extensive experiments on the BA-15 network to demonstrate that ResiNet can learn to optimize graphs with different resilience and utility metrics and to defend against other types of attacks besides the node degree-based attack, such as the node betweenness-based attack.

Table 3 records the improvements in percentage of ResiNet for varying objectives on the BA-15 dataset. As visualized in Figure 7, ResiNet is not limited to defend against the node degree-based attack (Figure 7 (b)-(j)) and also learns to defend against the betweenness-based attack (Figure 7 (k)-(s)). Total three resilience metrics are used, with $\mathcal{R}$ denoting the graph connectivity-based resilience metric, $\mathcal{SR}$ being the spectral radius and $\mathcal{SR}$ representing the algebraic connectivity. Total two utility metrics are adopted, including the global efficiency $E_{global}$ and the local efficiency $E_{local}$. Not surprisingly, the optimized network with an improvement of about 3.6% for defending the betweenness-based attack also has a higher resilience (around 7.8%) against the node-degree attack. This may be explained as the similarity between node degree and betweenness for a small network.

To balance utility and resilience simultaneously, the distribution of Pareto points obtained by ResiNet on the BA-15 dataset in Figure 8 implies that the maximization of the resilience and utility may not be satisfied simultaneously. The simultaneous improvements in resilience and utility could occur if the poorly designed original network has weak resilience and utility.

### D.2.2 INDUCTIVITY ON LARGER DATASETS

Even with limited computational resources, armed with the autoregressive action space and the power of FireGNN, ResiNet can be trained fully end-to-end on graphs with hundreds of nodes using RL. We demonstrate the inductive ability of ResiNet on graphs of different sizes by training ResiNet on the BA-20-200 dataset, which consists of graphs with the size ranging from 20 to 200, and then report its performance on directly guiding the edges selections on unseen test graphs. The filtration

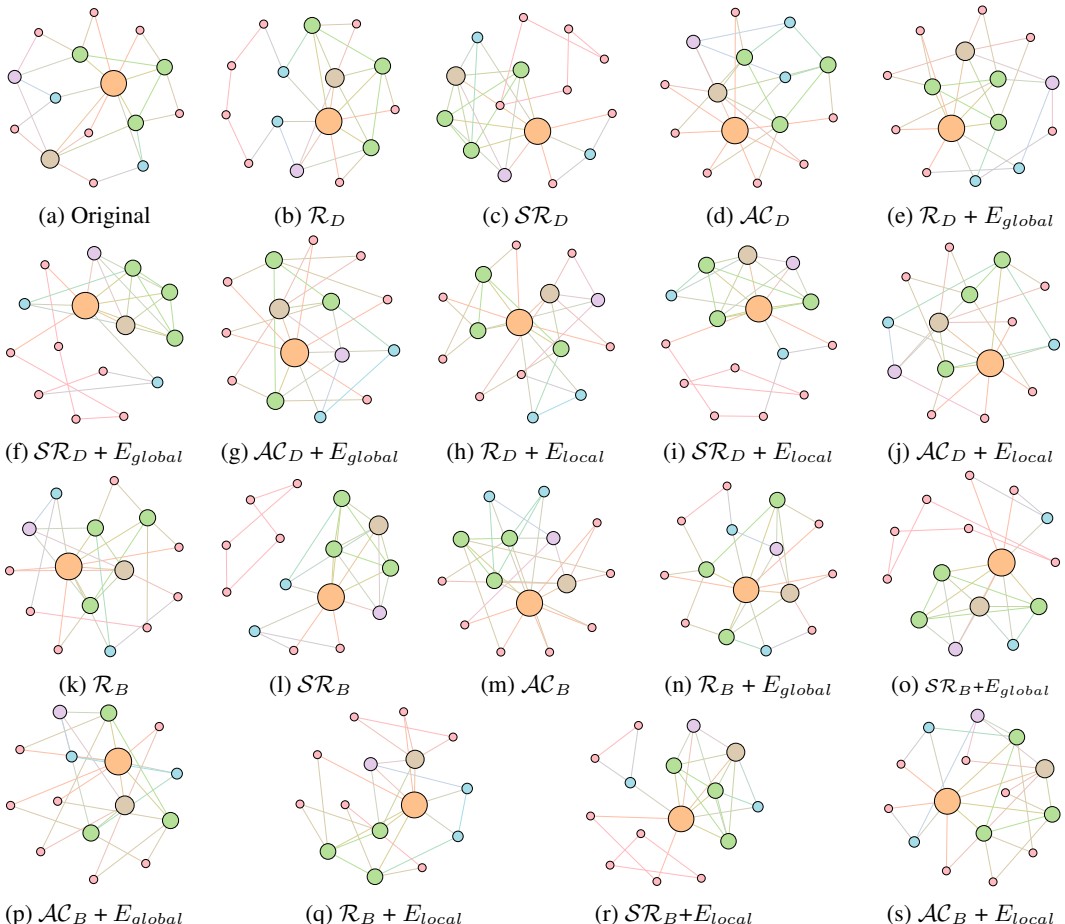

Figure 7: The resilience maximization on the BA-15 dataset with 15 nodes and 27 edges with (a) original network, (b)-(j) results of defending the node degree-based attack with different combinations of resilience and utility, and (k)-(s) results of defending against the node betweenness-based attack with varying combinations of resilience and utility. For three resilience metrics, $\mathcal{R}$ denotes the graph connectivity-based resilience metric; $\mathcal{SR}$ is the spectral radius; $\mathcal{SR}$ represents the algebraic connectivity. For two utility metrics, $E_{global}$ denotes the global efficiency, and $E_{local}$ is the local efficiency.

order $K$ is set to 1 for the computational limitation. As shown in Figure 9, we can see that ResiNet has the best performance for $N \in [70, 100]$. The degrading performance with the graph size may be explained by the fact that larger graphs require a larger filtration order for ResiNet to work well. A more stable performance improvement of ResiNet is observed with the increment of graph size when trained to optimize network resilience and utility simultaneously, and ResiNet possibly finds a strategy to balance these two metrics.

### D.2.3 INSPECTION OF OPTIMIZED NETWORKS

Moreover, to provide a deeper inspection into the optimized network structure, we take the EU power network as an example to visualize its network structure and the optimized networks given by ResiNet with different objectives. Comparing to the original EU network, Figure 10 (b) is the network structure obtained by only optimizing the graph connectivity-based resilience. We can observe a more crowded region on the left, consistent with the "onion-like" structure concluded in previous studies. If we consider the combination gain of both resilience and utility, we observe a more compact clustering "crescent moon"-like structure as shown in Figure 10 (c).

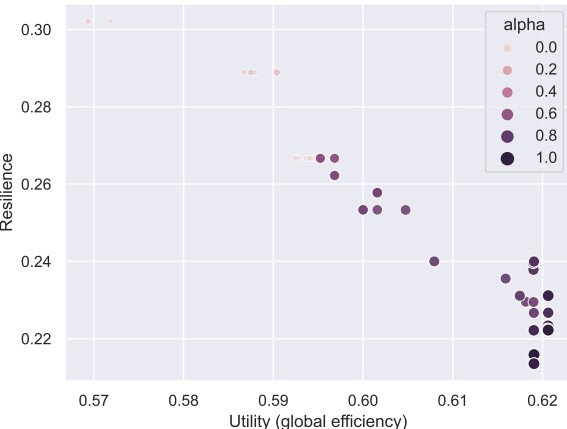

Figure 8: Pareto points obtained by ResiNet of balancing various combinations of resilience and utility on the BA-15 dataset. The resilience metric is the graph connectivity-based resilience, and the utility metric is the global efficiency. Results are averaged over three runs.

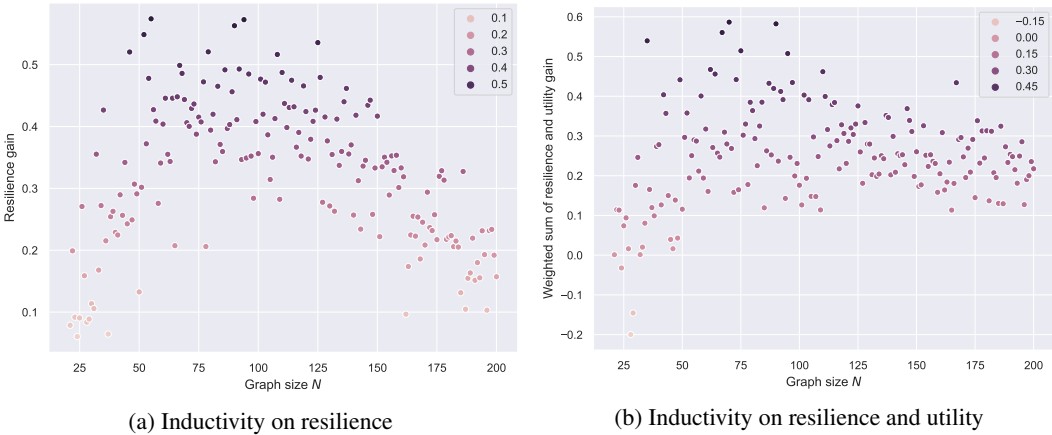

(a) Inductivity on resilience

(b) Inductivity on resilience and utility

Figure 9: The inductive ability of ResiNet on the test dataset (BA-20-200) when optimizing (a) network resilience and (b) the combination of resilience and utility.

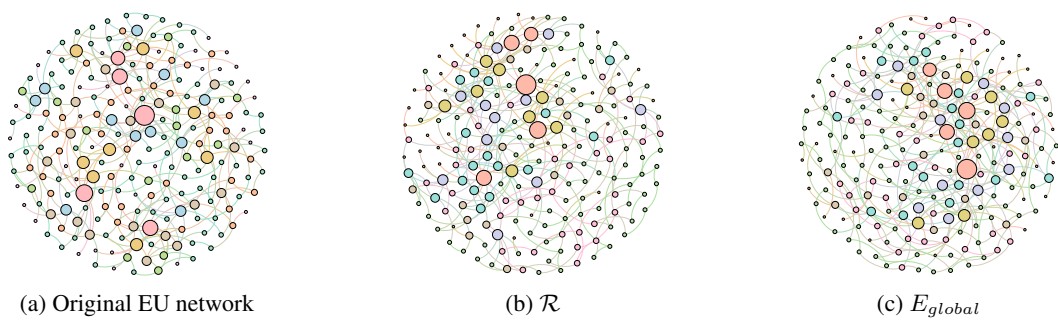

(a) Original EU network          (b) $\mathcal{R}$          (c) $E_{global}$

Figure 10: Visualizations of the original EU network and optimized networks using ResiNet with different objectives: $\mathcal{R}$ means the connectivity-based resilience measurement and $E_{global}$ is the global efficiency.

Table 4: Resilience optimization algorithm under the fixed maximal rewiring number budget of 200. We report the weighted combination of the graph connectivity-based resilience and the network efficiency improvement (in percentage) with the weighted coefficient $\alpha \in \{0, 0.5\}$. Results are averaged over 3 runs. We report the rewiring number inside the bracket.

| Method | $\alpha$ | BA-15 | BA-50 | BA-100 | EU |
|---|---|---|---|---|---|
| HC | 0 | 26.8 (10.0) | 52.1 (47.0) | 76.9 (97.3) | 71.9 (152.7) |
|  | 0.5 | 18.6 (11.3) | 43.1 (62.7) | 56.9 (121) | 63.2 (200) |
| SA | 0 | 26.8 (20) | 49.7 (59.0) | 84.5 (119.7) | 73.5 (160.3) |
|  | 0.5 | 17.8 (21) | 41.1 (79.7) | 57.7 (127.7) | 62.8 (200) |
| Greedy | 0 | 23.5 (6.0) | 48.6 (13.0) | \ | \ |
|  | 0.5 | 5.3 (15.0) | 34.7 (13.0) | \ | \ |
| EA | 0 | 35.3 (\) | 50.2 (\) | 61.9 (\) | 66.2 (\) |
|  | 0.5 | 27.1 (\) | 38.3 (\) | 46.6 (\) | 58.4 (\) |
| GNN+RL | 0 | 13.7 (2) | 0 (1) | 0 (1) | 9.0 (20) |
|  | 0.5 | 10.9 (2) | 0 (1) | 0 (1) | 2.1 (20) |
| ResiNet | 0 | 35.3 (6) | 61.5 (20) | 70.0 (20) | 54.2 (20) |
|  | 0.5 | 26.9 (20) | 53.9 (20) | 53.1 (20) | 51.8 (20) |

Table 5: Running speed (in second) of the resilience optimization algorithm under the fixed maximal rewiring number budget of 20 and 200 (in bracket). Results are averaged over 3 runs.

| Method | $\alpha$ | BA-15 | BA-50 | BA-100 | EU |
|---|---|---|---|---|---|
| HC | 0 | 1.0 (1.0) | 1.1 (6.4) | 1.3 (22.2) | 3.1 (94.2) |
|  | 0.5 | 1.5 (11.5) | 1.1 (12.8) | 2.0 (49.0) | 5.3 (193.7) |
| SA | 0 | 1.1 (1.1) | 0.3 (6.6) | 0.6 (22.6) | 2.4 (91.2) |
|  | 0.5 | 0.7 (1.7) | 0.7 (13.2) | 1.7 (47.5) | 5.0 (193.5) |
| Greedy | 0 | 0.2 (6.0) | 34.1 (34.5) | 766.3 (\) | 3061.7 (\) |
|  | 0.5 | 0.7 (0.7) | 64.1 (65.4) | 1478.9 (\) | 6192.6 (\) |
| EA | 0 | 0.01 (\) | 0.1 (\) | 1.59 (\) | 0.2 (\) |
|  | 0.5 | 0.01 (\) | 0.1 (\) | 0.8 (\) | 0.4 (\) |
| GNN+RL | 0 | 0.05 (\) | 0.1 (\) | 0.13 (\) | 3.6 (\) |
|  | 0.5 | 0.05 (\) | 0.1 (\) | 0.19 (\) | 4.5 (\) |
| ResiNet | 0 | 0.47 (\) | 1.83 (\) | 2.23 (\) | 4.5 (\) |
|  | 0.5 | 0.48 (\) | 1.89 (\) | 2.4 (\) | 5.2 (\) |

### D.2.4 PERFORMANCE COMPARISONS UNDER A LARGE REWIRING BUDGET

In this section, we present the resilience improvement and the required number of edge rewiring of each algorithm under a large rewiring budget of 20. The running speed is also presented to compare the running time efficiency of each algorithm.

As shown in Table 4, traditional methods improve the network resilience significantly compared to ResiNet under a large rewiring budget of 200. However, traditional methods are still undesired in such a case since a solution with a large rewiring budget is not applicable in practice due to the vast cost of adding many new edges into a real system. For example, the actual number of rewiring budget for EA is hard to calculate since it is a population-based algorithm, so it is omitted in Table 4. All baselines adopt the early-stopping strategy that they will terminate if there is no positive resilience gain in a successive 1000 steps.

Table 5 indicates that the time it takes for the benchmark algorithm to solve the problem usually increases as the test data set size increases. In contrast, once trained, our proposed ResiNet is suitable for testing on large datasets since the model size is fixed.

### D.2.5 LIMITATIONS

It should be noted that the weighted combination of resilience and utility setting can be slightly unfair for ResiNet due to the additional RL training burden. Despite our highly tuned implementation of ResiNet, training RL is known as challenging, with two reward functions with different ranges posing another challenge. Moreover, due to limited computational resources, we only train ResiNet with all non-empty subgraphs in FireGNN on the BA-15 network as a transductive and BA-10-30 network as an inductive setting. For networks with more than 50 nodes, we only use the top-2 subgraphs in the filtration, as we observed to balance the computational effort and the satisfying performance. This may explain the performance of ResiNet is slightly weaker than several baselines on larger networks.

Another point is that for ResiNet, the maximum episode length is set to 20 steps on all graphs, which means all solutions given by ResiNet require no longer than 20 edge rewiring operations. For example, the optimal solution obtained by ResiNet defending against node degree-based attacks on BA-15 only requires a minimum of six steps to arrive at the target graph. A larger filtration order, a longer episode length, or a powerful RL learning algorithm should be expected to boost the performance of ResiNet further.

Despite the success of ResiNet armed with our proposal FireGNN in both the transductive and inductive settings, we empirically observe that the trained agent only works well on the unseen datasets with a similar setting as training. How to improve further the inductivity of ResiNet is left as another future work.

