# OpenReview forum: "Edge Rewiring Goes Neural: Boosting Network Resilience via Policy Gradient"
_ICLR.cc/2022/Conference — ICLR 2022 Submitted_

### Official Review · Reviewer_Qtac · 2021-11-02

**Correctness:** 3
**Technical Novelty And Significance:** 2
**Empirical Novelty And Significance:** 2
**Recommendation:** 3
**Confidence:** 5

**Main Review:**

Positives:

+ The problem is relevant. In particular, the idea of neural approaches to increase network resilience is interesting.
+ The paper is easy to follow.

Negatives:

-  Method: The paper overclaims a bit. Among the three major points in the introduction (utility, local optimality, and transductivity); the last two are already addressed by previous methods that use reinforcement learning with graph neural networks (GNNs). Please refer to the survey [1] and especially [2]. It is also a little surprising to see the paper is not positioned in the context of those existing works that are very similar [1] in terms of frameworks. Given those, It is difficult to assess the additional technical novelty of the proposed method. The paper could be improved by focusing on specific challenges (such as scalability) that the rewiring problem has while working with this framework. The proposed method's efficiency is also unclear as there is no analysis of efficiency.

[1] Mazyavkina, Nina, Sergey Sviridov, Sergei Ivanov, and Evgeny Burnaev. "Reinforcement learning for combinatorial optimization: A survey." Computers & Operations Research (2021): 105400.
[2] Dai, Hanjun, Elias B. Khalil, Yuyu Zhang, Bistra Dilkina, and Le Song. "Learning combinatorial optimization algorithms over graphs." NeurIPS, 2017.


- Experiments: The experiments can be significantly improved.

a) The experiments have small datasets. Besides they are mostly synthetic. From Table 1, it is also difficult to say that the proposed method, ResiNet is a superior method.

b) There is no comparison with non-neural approaches in terms of efficiency. One of the major reasons for going to the neural version is to be able to test on big data while training on small ones. The neural approaches might be able to outperform the non-neural methods at least in terms of efficiency.

 c) There are no comparisons with other neural baselines that address graph combinatorial problems via neural approaches especially the ones which use reinforcement learning.

d) Why are the number of rewired edges not being compared among different methods? The paper claims that the proposed method uses a low number of edges to rewire to achieve high network resilience.



 - Motivation:
The motivation going into the neural version is not clear. Why will someone use a neural algorithm compared to the non-neural ones? The context where the algorithm needs to be used repetitively might motivate this in a better way (please refer to Khalil et al., NeurIPS, 2017). Could you provide other types of use cases?

**Summary Of The Paper:**

The paper proposes a neural approach that increases network resilience by edge wiring. The approach uses a combination of graph neural network (GNN) and policy gradient method to do so. The experiments use a few small networks and several non-neural baselines.

**Summary Of The Review:**

The novelty of the proposed method is unclear and the experiments need major improvement.

---

> ### Author Response · Authors · 2021-11-16
> **Response to Reviewer to Qtac (first part)**
>
> 1. > The paper overclaims a bit. Among the three major points in the introduction (utility, local optimality, and transductivity); the last two are already addressed by previous methods that use reinforcement learning with graph neural networks (GNNs).
>
>     For the resilience task we study in our paper, no current methods address the local optimality and transductivity to select two edges from the evolving dynamic graph, so it is a little hard for us to agree with the reviewer’s words “already addressed”. Moreover, we think that the reviewer may misunderstand our claim of transductivity. The transductivity of the edge rewire operation in our paper is not referred to using GNN to encode graphs with different permutations into the same latent vector.  The tranductivity that we mean is how to select edges from the graph inductively, which is challenging for the edge rewiring operation.  This is because the node degree sequence is not changing, and regular GNNs cannot generate distinguishable edge embeddings to let the RL agent successfully select different edges, which is the dilemma of the GNN’s expressive power that we discussed in the introduction (Page 2, fourth paragraph).
>
>     In fact, we never claim we are the first to use the GNN and RL to solve tasks defined on graphs, and we also discussed these studies in the Section Extend Related Work (Appendix A, Page 14).  Although the previous work uses GNN and RL to solve some tasks on graphs, it does not mean all tasks on graphs can be easily addressed using the GNN+RL framework. As we clearly emphasize in both abstract and introduction (Page 2), our proposed method overcomes these three limitations of current methods for optimizing network resilience, which is challenging to select two distinct edges from the evolving graph at each step. To the best of our knowledge, no current work successfully addresses these three limitations for optimizing network resilience.  We believe the reviewer should first realize the difference between the network resilience task (two edges-level prediction on an evolving dynamic graph)  and well-studied CO problems like TSP or MVC (node-level prediction on a static input graph). The degree-preserving edge rewiring process makes it challenging to directly adapt regular GNN and RL to solve it without ResiNet.
>
> 2. >  It is also a little surprising to see the paper is not positioned in the context of those existing works that are very similar [1] in terms of frameworks. Given those, it is difficult to assess the additional technical novelty of the proposed method.
>
>      Thanks for the reviewer pointing it out. In fact, we already discussed the related work using GNN for combinatorial problems in Appendix A (Page 14) due to the page limit of the main text.  To avoid more confusion, we move the discussion of the related work on using GNN for CO to the main text.  In fact, in the paper, we have clearly discussed our technical contributions, such as FireGNN (the introduction and Sec 3.3.1) and the permutation-invariant edge selection model. We observe the review does not include any feedback about the FireGNN. It seems that the reviewer may overlook our proposed FireGNN, which is the essential component to make GNN and RL work on the resilience task. Therefore we hope the reviewer can go through the technical part of the paper with patience before the eagerness to regard our work as a simple combination of GNN and RL.
>
>     We have discussed that replacing FireGNN with regular GNNs cannot learn the edge rewiring operation in Appendix C.3 (Page 18). However, as suggested by the reviewer, we realize that we need to demonstrate our ResiNet's advantage better. Therefore we add the ablation study on FireGNN in Table 1 (Page 8) to compare our proposed method to the traditional method using a regular GNN with the RL framework as a neural version baseline.  We hope this more formal ablation study is enough to prove the technical novelty of our proposed ResiNet and highlight the difference between our algorithm and existing algorithms since it would be impossible to use regular GNN and RL to solve this problem.

---

> > ### Author Response · Authors · 2021-11-16
> > **Response to Reviewer to Qtac (second part)**
> >
> > 3. > The experiments can be significantly improved. From Table 1, it is also difficult to say that the proposed method, ResiNet is a superior method.
> >
> >    We apologize for the misunderstanding of the performance comparison in Table 1. The empirical performance comparison in the initial paper is not presented correctly. We carefully revised our paper to indicate better that our ResiNet outperforms all baselines consistently on all datasets.
> >
> >    In the initial version, the reason why our proposed method performs much worse than baselines on large graphs is two-fold. First, all baselines optimize the resilience of the input graph iteratively, and we did not set a limitation of maximum valid edge rewiring operations for them. Baselines can rewire hundreds of times to obtain a significant resilience gain, which would be costly in practice since each rewiring introduces a new cost. At the same time, our ResiNetis limited to a maximum of 20 steps since we empirically found that training ResiNet with a longer episode length leads to no extra gain. This may be due to the general hardness of training RL under the degree-preserving constraint.   For graphs with sizes 100 and 200, it leads to an ample action space in RL. This requires a large amount of computing resources to reach its full potential, which is a common factor affecting RL-based approaches' performance.
> >
> >    To make the solution given by different algorithms applicable in practice, in the revised version, we set a maximum rewiring budget of 20, so updating existing real systems will not be too costly.  As we present in Table 1 (Page 9), our proposed approach consistently outperforms all baselines on graphs with different sizes. We hope our work can inspire more studies to focus on this critical problem.
> >
> >    The table (Table 1 in the paper ) below records the revised experimental results under the maximal rewiring number budget of 20, including the resilience gain and the required number of the edge rewiring (in the bracket).
> >
> >     |         	| alpha 	| BA-15       	| BA-50     	| BA-100    	| EU        	|
> >     |---------	|-------	|-------------	|-----------	|-----------	|-----------	|
> >     |    HC   	| 0     	| 26.8 (10)   	| 30.0 (20) 	| 24.1 (20) 	| 19.8 (20) 	|
> >     | HC      	| 0.5   	| 18.6 (11.3) 	| 22.1 (20) 	| 14.9 (20) 	| 16.3 (20) 	|
> >     | SA      	| 0     	| 21.6 (17.3) 	| 11.9 (20) 	| 12.5 (20) 	| 14.9 (20) 	|
> >     | SA      	| 0.5   	| 16.8 (19.0) 	| 11.4 (20) 	| 13.4 (20) 	| 14.0 (20) 	|
> >     | Greedy  	| 0     	| 23.5 (6)    	| 48.6 (13) 	| 64.3 (20  	| 0.5 (3)   	|
> >     | Greedy  	| 0.5   	| 5.3 (15)    	| 34.7 (13) 	| 42.7 (20) 	| 0.3 (3)   	|
> >     | EA      	| 0     	| 8.5 (20)    	| 6.4 (20)  	| 4.0 (20)  	| 8.2 (20)  	|
> >     | EA      	| 0.5   	| 6.4 (20)    	| 4.7 (20)  	| 2.8 (20)  	| 9.3 (20)  	|
> >     | GNN+RL  	| 0     	| 13.7 (2)    	| 0 (1)     	| 0 (1)     	| 9.0 (20)  	|
> >     | GNN+RL  	| 0.5   	| 10.9 (2)    	| 0 (1)     	| 0 (1)     	| 2.1 (20)  	|
> >     | ResiNet 	| 0     	| 35.3 (6)    	| 61.5 (20) 	| 70.0 (20) 	| 54.2 (20) 	|
> >     | ResiNet 	| 0.5   	| 26.9 (20)   	| 53.9 (20) 	| 53.1 (20) 	| 51.8 (20) 	|
> >
> >
> > 4. >  The experiments have small datasets.
> >
> >    On the one hand, the datasets used in our experiments are not large in size since the maximal number of nodes is 200.  However, it should be noted that the graph size is not the only metric to determine one problem’s complexity. For RL-based solutions, the action space size is a more appropriate metric to determine the complexity.  For example, the problem scale for a graph with hundreds of nodes may be small for many node-level prediction CO problems. The node-level prediction means that methods usually need to select a node to add to the solution set at each step, so the complexity is O(N) at each step. However,  we want to point out that the dataset size we used is not small for the resilience task via the edge rewiring we study since the problem has an ample action space for graphs with hundreds of nodes.  This is because, for the edge rewiring operation, we need to select two different edges from the graph, and therefore the time complexity is O(N^4) at each step. This implies that most transductive methods only efficiently solve the task on graphs of hundreds or thousands of nodes. The maximum dataset size we used in the paper is a BA network with 200 nodes and 792 edges. The theoretical maximum action space size is 1254528, corresponding to a significant problem size for other node-level CO problems like TSP.
> >
> > 5. > Besides they are mostly synthetic.
> >
> >     As we describe in Appendix C.1 (Page 17), the BA network is commonly used to test the performance of an algorithm on defending the malicious attacks on graphs since they are vulnerable to malicious attacks on networks and are commonly used to test network resilience optimization algorithms. Moreover, we test our proposed method on both synthetic and the EU power network for both inductive and transductive settings.

---

> > > ### Author Response · Authors · 2021-11-16
> > > **Response to Reviewer to Qtac (third part)**
> > >
> > > 6. >   There is no comparison with non-neural approaches in terms of efficiency. One of the major reasons for going to the neural version is to be able to test on big data while training on small ones. The neural approaches might be able to outperform the non-neural methods at least in terms of efficiency.
> > >
> > >    The resilience objective is time-consuming to calculate, especially for large graphs, which is the main reason why current network resilience optimizations methods cannot be scaled well to large graphs with more than thousands of nodes. Our proposed method can directly generate the solution without querying the resilience and without the try-and-trial search process. Therefore, the efficiency advantage of neural approaches is obvious compared to non-neural methods on improving the resilience of new graphs even if there is no comparison with non-neural approaches in terms of efficiency.  However,  to make it more straightforward and convincing,  as suggested by the reviewer, a detailed comparison with all methods in terms of efficiency is added in Table 5 (Page 22).  Since the model capacity of ResiNet is fixed, once trained, ResiNet can efficiently be applied to optimize graphs with different sizes.
> > >
> > > 7. > There are no comparisons with other neural baselines that address graph combinatorial problems via neural approaches especially the ones which use reinforcement learning.
> > >
> > >      It seems that a simple combination of GNN and RL could be used on the resilience task. However, as we have explained before, a baseline, the simple combination of GNN and RL, we added in Table 1 by replacing our FireGNN with a regular GNN model, cannot work.
> > >
> > >     Due to the constraint of preserving node degree, to our best knowledge, no deep graph generation work can control the node degree of the generated graphs without modeling edge rewiring as an MDP process as we did.
> > >
> > > 8. >  Why are the number of rewired edges not being compared among different methods? The paper claims that the proposed method uses a low number of edges to rewire to achieve high network resilience.
> > >
> > >      As suggested by the reviewer, we compare the number of rewired edges among different methods. And the result shown in Table 1 validates our claim that the proposed method uses a low number of edges to rewire to achieve high network resilience for small graphs.
> > >
> > > 9. > The motivation going into the neural version is not clear. Why will someone use a neural algorithm compared to the non-neural ones? The context where the algorithm needs to be used repetitively might motivate this in a better way (please refer to Khalil et al., NeurIPS, 2017). Could you provide other types of use cases?
> > >
> > >     (1) The first motivation for the neural version is to overcome the local optimality issue of current baselines based on RL.
> > >
> > >     (2) The second motivation is that the neural version implies that a trained ResiNet can directly optimize unseen graphs without the search process.  In the real world, lots of systems need to improve resilience via edge rewirings. Traditional methods solve each graph one by one, which is time-consuming for large graphs.
> > >
> > >     (3) For the setting of handling dynamic graphs, like the rapidly changing social networks and communication networks, a small network structure change requires baselines to be executed from scratch, which means the algorithm needs to be used repetitively.
> > > Thanks to the reviewer for pointing this out.

---

> > > > ### Comment · Reviewer_Qtac · 2021-11-29
> > > > **Thanks for the response**
> > > >
> > > >
> > > > Thank you for your response. Some of my major concerns still are scalability (and thus the applicability of the proposed method) as well as the baselines. For the scalability part, rigorous computations of training/inference times, use of large datasets, changing some components of the architecture might help. At this point, I am unable to update my score.

---

> > > > > ### Author Response · Authors · 2021-11-30
> > > > > **Thanks for the reviewer's response and Further clarification from the authors**
> > > > >
> > > > > Thanks for the response. We appreciate that our previous comments have addressed the most concerns of the reviewer.  Furthermore, we would like to explain further about the rest concerns.
> > > > >
> > > > > > scalability
> > > > >
> > > > > The scalability should be an advantage of our proposed method instead of a drawback. Since we still managed to learn an agent to optimize the resilience of different graphs inductively, despite the challenge of the edge rewiring.  Two factors prevent existing methods from being scaled to larger datasets for the resilience maximization task via edge rewiring. The first one is the need to select two edges from an evolving graph at each step which has a complexity of $O(N^4)$.  The second one is the time-consuming reward calculation. For example,  the resilience value based on the spectrum-based measurement is time-consuming to calculate frequently for large graphs.  Therefore, as we have explained about the dataset size in the fourth comment above, we would like to clarify again. **The graph size is not the proper way to judge the scalability of an algorithm**. **It is the decision space (action space in RL) that influences the scalability of an algorithm**.   We presented the action space size of each dataset in Table 2 (Page 17 in the revision) as follows.
> > > > >
> > > > > | Dataset  | Node | Edge | Action Space Size |
> > > > > | ------------- | ------------- | ------------- | ------------- |
> > > > > | BA-15 | 15  | 54   | 5832
> > > > > | BA-50 | 50  |   192  |73728
> > > > > | BA-100| 100    |    392 | 307328
> > > > > | EU  |  217  |   640   | 819200
> > > > > | BA-10-30  | 10-30   |  112   | 25088
> > > > > | BA-20-200  |  20-200  |  792    | 1254528
> > > > >
> > > > > The above table shows why the problem is challenging even for graphs of hundreds of nodes since the decision space is tremendous (million).
> > > > >
> > > > > Our paper does not focus on the excellent scalability of the model so it can optimize large more networks than baselines. Without the invention of our proposed FireGNN as a refinement of standard GNNs,  a combination of GNN and RL cannot work on graphs with ten nodes. In contrast, our method can work thanks to the FireGNN,  optimize networks with hundreds of nodes and inductively boost the resilience of different networks.
> > > > >
> > > > > As the first work to successfully optimize the resilience of networks up to hundreds of nodes with inductivity, scalability is not a drawback of our method. More computational resources may help more, but it is not a focus of our paper and can be left as future work. If the reviewer is still concerned about scalability, we would like to know the expected acceptable dataset size and corresponding decision space size.
> > > > >
> > > > > > baselines.
> > > > >
> > > > > We believe that we have clearly explained about the baselines part in the 7-th comment above, and we have already added the combination of regular GNN and RL as a neural-version baseline.  To the best of our knowledge, there is no neural-version baseline that can solve the resilience task via edge rewiring, not to mention that our work can inductively optimize the resiliences of different networks.   If the reviewer is still concerned about the baseline part, could the reviewer provide some other neural-version baselines that can also boost the resilience via edge rewirings which we can add to the paper? Thanks!
> > > > >
> > > > > > At this point, I am unable to update my score.
> > > > >
> > > > > We feel a little disappointed about the score and the variance compared to second reviewer's positive recommendation, since our proposed method as an initial neural-version study about the resilience task can inspire more work on this problem.  **We are the first to propose a novel GNN variant to study this resilience problem, which bridges another gap between the network science field and the machine learning field and should be encouraged**.  Most current CO problems focus on constructing the node-level solution on a static input graph instead of the edge-level solution on dynamic evolving graphs. Similarly, the PointerNet [1] was the first to solve the TSP problem, and it only can solve no more than 50 nodes for a TSP task with $O(N)$ complexity. In contrast, our method can solve hundreds of nodes for the resilience task via edge rewiring with $O(N^4)$ complexity.    Although PointerNet was proposed in 2015 and current methods for TSP can solve graphs with million nodes. The situation is entirely different for the resilience task we study in the paper.  **The decision space size of the dataset we used in the paper is already million for the resilience task, which is comparable to SOTAs for TSP**.  We spent plenty of time finetuning the implementation to let the model work on hundreds of nodes inductively since the problem is challenging. We have open-sourced our proposed model to wish to benefit the community.
> > > > >
> > > > > [1] Vinyals, Oriol, Meire Fortunato, and Navdeep Jaitly. "Pointer networks." Advances in Neural Information Processing Systems (2015).
> > > > >
> > > > > We thank the reviewer for the time and effort. We hope we have addressed all concerns of the reviewer, including the scalability and baselines.

---

> > > > > > ### Comment · Reviewer_Qtac · 2021-11-30
> > > > > > **Thanks for the response**
> > > > > >
> > > > > > Thanks for the effort, again! I do not see how the concerns are getting addressed without improving the architecture and running the experiments on large graphs. However, I really hope that these comments from all the reviewers will be taken positively and the paper can be improved significantly. For baselines, please refer to this recent survey [1].
> > > > > >
> > > > > > [1]  Nina Mazyavkina, Sergey Sviridov, Sergei Ivanov, and Evgeny Burnaev. "Reinforcement learning for combinatorial optimization: A survey." Computers & Operations Research (2021).

---

> > > > > > > ### Author Response · Authors · 2021-11-30
> > > > > > > **Thanks for the response**
> > > > > > >
> > > > > > > We thank the reviewer very much for the prompt reply.
> > > > > > >
> > > > > > > > 1. Large graphs
> > > > > > >
> > > > > > > Despite that the reviewer maintains the claim that the dataset size used is small, it is not the case for solving CO problems with machine learning methods. In fact, state-of-the-art methods in TSP [1, 3, 6, 8] have a node size (action space size) of around more than ten thousands, methods in MVC [2, 3, 4] have a node size (action space size) of around hundreds of nodes and methods in VRP [5, 7, 9, 10] have a node size (action space size) of around hundreds of nodes . In addition, these works only require node-level prediction (linear complexity) while resilience optimization and edge rewiring requires edge-level prediction (quartic complexity). To the best of our knowledge, general machine learning methods for CO is still focusing on this scale of the problem.
> > > > > > >
> > > > > > > > 2. The survey
> > > > > > >
> > > > > > > We have added the discussion of the survey and the methods. Methods in it are relevant to CO problems, while none of them could be adapted to resilience optimization where the degree sequence needs to be maintained. In fact, all tasks included in this survey (TSP, MC, bin packing, MVC, MIS) are node-level predictions and extending them to resilience optimization will require the very techniques proposed in our manuscript. We believe the discussion will address the concerns of comparing our method to the methods in the survey.
> > > > > > >
> > > > > > > If some methods in the survey work on resilience optimization, could the reviewer please specify one?
> > > > > > >
> > > > > > > ***
> > > > > > > [1] Fu, Zhang-Hua, Kai-Bin Qiu, and Hongyuan Zha. "Generalize a Small Pre-trained Model to Arbitrarily Large TSP Instances." AAAI 2020.
> > > > > > >
> > > > > > > [2] Song, Jialin, et al. "Co-training for policy learning." Uncertainty in Artificial Intelligence. PMLR, 2020.
> > > > > > >
> > > > > > > [3] Dai, Hanjun, et al. "Learning combinatorial optimization algorithms over graphs." NeurIPS, 2017.
> > > > > > >
> > > > > > > [4] Manchanda, Sahil, et al. "Learning heuristics over large graphs via deep reinforcement learning." NeurIPS 2020.
> > > > > > >
> > > > > > > [5]  M. Nazari, A. Oroojlooy, L. Snyder, and M. Takác. Reinforcement learning for solving the vehicle routing problem. In Proceedings of the 32nd Conference on Advances in Neural Information Processing Systems, NeurIPS, pages 9839–9849, 2018.
> > > > > > >
> > > > > > > [6] M. Deudon, P. Cournut, A. Lacoste, Y. Adulyasak, and L.-M. Rousseau. Learning heuristics for the TSP by policy gradient. In Lecture Notes in Computer Science (including subseries Lecture Notes in Artificial Intelligence and Lecture Notes in Bioinformatics), 2018.
> > > > > > >
> > > > > > > [7] W. Kool, H. van Hoof, and M. Welling. Attention, learn to solve routing problems! In Proceedings of the 7th International Conference on Learning Representations, ICLR, 2019.
> > > > > > >
> > > > > > > [8] Q. Cappart, T. Moisan, L.-M. Rousseau, I. Prémont-Schwarz, and A. Cire. Combining reinforcement learning and constraint programming for combinatorial optimization. AAAI, 2021.
> > > > > > >
> > > > > > > [9] X. Chen and Y. Tian. Learning to perform local rewriting for combinatorial optimization. In Proceedings of the 33rd Conference on Advances in Neural Information Processing Systems, NeurIPS, pages 6281–6292, 2019.
> > > > > > >
> > > > > > > [10] H. Lu, X. Zhang, and S. Yang. A learning-based iterative method for solving vehicle routing problems. In International Conference on Learning Representations, 2020.

---

### Official Review · Reviewer_vtoA · 2021-11-03

**Correctness:** 3
**Technical Novelty And Significance:** 3
**Empirical Novelty And Significance:** 2
**Recommendation:** 6
**Confidence:** 3

**Details Of Ethics Concerns:**

N.A.

**Main Review:**

Strengths:

1. The proposed reinforcement learning based framework is suitable for the rewiring task. Meanwhile, the framework can generalize to unseen graphs since the edge rewiring policy can be directly applied to other graphs.

Weaknesses:

1. My main concern is the empirical performance of the proposed framework. Specifically, the proposed framework performs much worse than baseline approaches on large graphs (BA-100 dataset, EU dataset) as shown in Table 1. Also, the improvements on BA-15 dataset and BA-50 dataset are not significant.

Questions and suggestions:

1. I suggest the authors put the ablation study on the FireGNN and the new framework. For example, what is the performance of ResiNet using a standard GNN architecture (not using FireGNN)? What is the performance of replacing standard GNN with FireGNN in the previous method (not using the RL-based framework ResiNet)? These may help better understand the role of the architecture and the RL-based framework.

2. As mentioned in Section 4.2 (Page 8), 'Note that this comparable performance is under a much fewer number of rewiring operations.' Could the authors provide the number of rewiring operations of each method in Table 1?

3. As mentioned in Section 3.3.1 (Page 6), 'where $G^{(k)}$ denotes the remaining graph after removing $N-k$ nodes in a particular task-dependent order'. Could the authors clarify how the task-dependent order is determined?

Typo:

1. $d_i = \sum_{i=1}^{N}A_{ij}$ in Section 3.1 (Page 4).

**Summary Of The Paper:**

This paper studies how to improve network resilience by proposing a reinforcement learning-based framework named **ResiNet** and a new GNN architecture called **FireGNN**. The proposed framework is able to directly generalize to unseen graphs. The new GNN architecture applies the graph filtration process, which enhances the expressivity of GNN. The authors conduct experiments on synthetic and real datasets to compare the proposed framework with previous baseline methods.

**Summary Of The Review:**

This paper proposes a new RL-based framework together with a new GNN architecture for improving network resilience. The new frameworks can generalize to unseen graphs. However, the empirical performance of the framework is not convincing on the tasks considered in this paper.


=============================after response=============================

The updated results in the revision resolve some of my main concerns. I decide to raise my recommendation score from 5 to 6.

---

> ### Author Response · Authors · 2021-11-16
> **Response to Reviewer vtoA (first part)**
>
> We first thank the reviewer for their constructive review and feedback that improve the quality of our work.
>
> 1. > My main concern is the empirical performance of the proposed framework. Specifically, the proposed framework performs much worse than baseline approaches on large graphs (BA-100 dataset, EU dataset) as shown in Table 1. Also, the improvements on BA-15 dataset and BA-50 dataset are not significant.
>
>     The empirical performance comparison in the initial paper is not properly presented, and we carefully revised our paper to better present the performance improvement as follows:
>
>     In the initial version, the reason why our proposed method performs much worse than baselines on large graphs is two-fold. First, all baselines optimize the resilience of the input graph iteratively, and we did not set a limitation of maximum valid edge rewiring operations for them. Baselines can rewire hundreds of times to obtain a significant resilience gain, which would be costly in practice since each rewiring introduces a new cost. At the same time, our proposed method is limited to a maximum of 20 steps since we empirically found that training ResiNet with a longer episode length leads to no extra gain. This may be due to the general hardness of training RL under the degree-preserving constraint.   For graphs with sizes 100 and 200, it leads to an ample action space in RL. This requires a large amount of computing resources to reach its full potential, which is a common factor affecting RL-based approaches' performance.
>
>     To make the solution given by different algorithms applicable in practice, in the revised version, we set a maximum rewiring budget of 20, so updating existing real systems will not be too costly.  As presented in Table 1 (Page 9), our proposed approach outperforms all baselines on graphs with different sizes. We hope our work can inspire more studies to focus on this critical problem.
>
>     For the BA-15 dataset, our proposed method already achieved the optimal result, so that the improvement may look insignificant.
>
>
> 2. >  I suggest the authors put the ablation study on the FireGNN and the new framework. For example, what is the performance of ResiNet using a standard GNN architecture (not using FireGNN)? What is the performance of replacing standard GNN with FireGNN in the previous method (not using the RL-based framework ResiNet)? These may help better understand the role of the architecture and the RL-based framework.
>
>     We are thankful for the reviewer’s suggestion to better present the role of ResiNet and FireGNN.
>
>     However, we want to explain two things.
>
>    (1) FireGNN will be a standard GNN if the filtration order is set to 0, which means only the original input graph is passed into a GNN model.
>    **In the initial version**, the ablation study on the FireGNN is already included in Appendix Sec C.4 (Page 18) and C.5.1(Page 19), where we discussed the weak performance of ResiNet without the FireGNN on different datasets and provided several possible explanations why FireGNN is useful.  **In the revised version**, this study is included in Appendix Sec D.1.  In the initial version, we did not add the ablation study in Table 1 because replacing FireGNN with a regular GNN will not work. As suggested by the reviewer, to make the advantage of ResiNet and the role of FireGNN more clear, we include the result of the ablation study on FireGNN in the revised paper (please see Table 1, the ablation baseline is named as GNN+RL).
>
>     (2)  The ablation study on the new framework will be helpful. However, conducting the ablation study on the new framework regarding the edge rewiring operation is nontrivial. To the best of our knowledge, the RL framework we used is the only deep learning-based way to provide a solution for the network resilience maximization task while preserving the node degree. It is hard to use other frameworks to do the edge rewiring operation inductively. For example, current deep graph generation models cannot accurately control the node degree, which is undesired for leaning the edge rewiring operation. Therefore, the ablation study on the new framework is not considered.
>
>
> 3. > As mentioned in Section 4.2 (Page 8), 'Note that this comparable performance is under a much fewer number of rewiring operations.' Could the authors provide the number of rewiring operations of each method in Table 1?
>
>    We provide the number of rewiring operations of each method in Table 1 and compare different algorithms to show that our proposed method requires less number of rewirings to improve the network resilience significantly compared to baselines.

---

> > ### Author Response · Authors · 2021-11-16
> > **Response to Reviewer vtoA (second part)**
> >
> > 4. >  As mentioned in Section 3.3.1 (Page 6), 'where G(k) denotes the remaining graph after removing N−k nodes in a particular task-dependent order'. Could the authors clarify how the task-dependent order is determined?
> >
> >     For the resilience task that we study in the paper, as discussed in Appendix Sec C.3 (Page 18, initial paper), we use the node-degree centrality as an order to remove N-k nodes and construct the filtration.  To make it clear,  in the revised paper, we discuss the filtration order in a separate paragraph in Sec D.1 (page 18, revised paper).
> >
> >
> > 5.  > Typo
> >
> >      We correct the typo as the reviewer suggests. Thanks for pointing it out.
> > ***
> >  The table (Table 1 in the paper ) below records the revised experimental results under the maximal rewiring number budget of 20, including the resilience gain and the required number of the edge rewiring (in the bracket).
> >
> >    |         	| alpha 	| BA-15       	| BA-50     	| BA-100    	| EU        	|
> >    |---------	|-------	|-------------	|-----------	|-----------	|-----------	|
> >    |    HC   	| 0     	| 26.8 (10)   	| 30.0 (20) 	| 24.1 (20) 	| 19.8 (20) 	|
> >    | HC      	| 0.5   	| 18.6 (11.3) 	| 22.1 (20) 	| 14.9 (20) 	| 16.3 (20) 	|
> >    | SA      	| 0     	| 21.6 (17.3) 	| 11.9 (20) 	| 12.5 (20) 	| 14.9 (20) 	|
> >    | SA      	| 0.5   	| 16.8 (19.0) 	| 11.4 (20) 	| 13.4 (20) 	| 14.0 (20) 	|
> >    | Greedy  	| 0     	| 23.5 (6)    	| 48.6 (13) 	| 64.3 (20  	| 0.5 (3)   	|
> >    | Greedy  	| 0.5   	| 5.3 (15)    	| 34.7 (13) 	| 42.7 (20) 	| 0.3 (3)   	|
> >    | EA      	| 0     	| 8.5 (20)    	| 6.4 (20)  	| 4.0 (20)  	| 8.2 (20)  	|
> >    | EA      	| 0.5   	| 6.4 (20)    	| 4.7 (20)  	| 2.8 (20)  	| 9.3 (20)  	|
> >    | GNN+RL  	| 0     	| 13.7 (2)    	| 0 (1)     	| 0 (1)     	| 9.0 (20)  	|
> >    | GNN+RL  	| 0.5   	| 10.9 (2)    	| 0 (1)     	| 0 (1)     	| 2.1 (20)  	|
> >    | ResiNet 	| 0     	| 35.3 (6)    	| 61.5 (20) 	| 70.0 (20) 	| 54.2 (20) 	|
> >    | ResiNet 	| 0.5   	| 26.9 (20)   	| 53.9 (20) 	| 53.1 (20) 	| 51.8 (20) 	|

---

> > > ### Comment · Reviewer_vtoA · 2021-11-27
> > > **Thanks for the response**
> > >
> > > Thank you for your response and the updated results. Overall, the updated results (e.g., table 1) demonstrate the effectiveness of the proposed method, and I decide to raise my recommendation score to 6.
> > >
> > > Also, the section C.5.1 is missing in the current revision.

---

> > > > ### Author Response · Authors · 2021-11-28
> > > > **Thanks for the reviewer's response**
> > > >
> > > > We appreciate your constructive suggestions to make our work better.
> > > >
> > > > As for C.5.1, we are sorry for this misunderstanding.  C.5.1 is referred to the initial version of the paper, and the corresponding part of the ablation study on FireGNN is changed to Appendix D.1 in the revision.  Thanks for pointing it out, and we have added the version information in the comment.
> > > >
> > > > We sincerely thank the reviewer for the suggestions again.

---

### Official Review · Reviewer_AvkM · 2021-11-04

**Correctness:** 4
**Technical Novelty And Significance:** 1
**Empirical Novelty And Significance:** 1
**Recommendation:** 3
**Confidence:** 3

**Main Review:**

Strengths

a new approach based on neural networks.

Weaknesses

The main motivation is to improve the robustness of infrastructure networks such as power networks.  While rewiring an edge sounds straightforward on a graph, it is tremendously difficult in a physical network, e.g. adding a new road or a new power line. Furthermore, the functionality of a physical network is beyond nodes and links. For example, for a power network, power flow is determined by power flow equations. So considering the graph structure alone, without the dynamics of the systems, is not sufficient.

**Summary Of The Paper:**

The paper proposes a neural network based algorithm for edge rewiring.

**Summary Of The Review:**

The problem is not very motivated.

---

> ### Author Response · Authors · 2021-11-16
> **Response to Reviewer AvkM**
>
> Based on the comments given by the first reviewer, which is entirely different from the feedbacks given by other reviewers, we realize that the reviewer may have looked into the problem from the perspective of practical power systems. The detail of the power system is essential, but it is not the focus of our paper. It is not necessary that every study focuses on the implementations of such systems. It also does not attract too much attention in machine learning and network science.
>
> Regardless, we provide detailed explanations for all questions given by the reviewer. **Based on the explanations below, we sincerely hope that the reviewer can evaluate the novelty of our work by focusing on improving network resilience in the field of network science and machine learning.**
> 1. >  The problem is not very motivated.
>
>    We are surprised that the reviewer thinks that improving network resilience by edge rewiring is not motivating since numerous network science studies focused on this crucial topic. Some of them are also cited in our work.
>
>    [1] Schneider, Christian M., André A. Moreira, José S. Andrade, Shlomo Havlin, and Hans J. Herrmann. "Mitigation of malicious attacks on networks." Proceedings of the National Academy of Sciences 108, no. 10 (2011): 3838-3841.
>
>    [2] Gao, J., Buldyrev, S., Stanley, H. et al. Networks formed from interdependent networks. Nature Physics, 8, 40–48 (2012).
>
>    [3] Zhou, Qiong, and Janusz W. Bialek. "Approximate model of European interconnected system as a benchmark system to study effects of cross-border trades." IEEE Transactions on Power Systems, 20, no. 2 (2005): 782-788.
>
>
> 2. > So considering the graph structure alone, without the dynamics of the systems, is not sufficient.
>
>    The reviewer thinks considering the graph structure alone without the dynamics of the systems is not sufficient. First, in the field of network science, as cited above, a network is a commonly-used mathematical tool to represent and analyze the resilience of  an infrastructure system. Improving the network resilience by modeling the system as a graph is based on tremendous well-studied papers.  Second,  the power network only serves as an application example to show that many systems are vulnerable to malicious attacks.  Our algorithm is applicable to many systems that can be represented as a graph.
>
>
> 3. > It is tremendously difficult in a physical network,  e.g. adding a new road or a new power line.
>
>    Regarding this question, it is indeed costly to add a new road or a new power line. However, it should be noted that losing the whole system due to malicious attacks or natural failures is more costly than adding a new power line. It is the duty of the administrator or the local government to balance the benefit of improving the network resilience, the cost of adding new edges, and the cost of losing the whole system functionality. If a poorly-designed network system is vulnerable to malicious attacks, improving its resilience at some cost is sensible even if it costs some resources. Finally, our method is not limited to the power networks. There are still many networks that need to improve their resilience, such as social networks, computer communication networks.
>
>    We also want to point out that this cost issue validates our method's advantage as our model requires less number of rewirings to optimize the network resilience than baselines.

---

### Author Response · Authors · 2021-11-19
**Revision Summary**

We thank all reviewers for the careful and constructive reviews. We revised the paper accordingly and marked the major modifications in blue for visibility. The major revision changes are summarized as follows.
- We have updated the experimental results to show that our proposed method consistently outperforms all baselines under the same rewiring budget in Table 1 (vtoA & Qtac).
- We have added the ablation of FireGNN as a neural version baseline to more clearly demonstrate the advantage of our proposed method in Table 1. We also have added the discussion of the critical role of FireGNN in Appendix D.1 (vtoA & Qtac)
- We have added the setup detail of FireGNN in Appendix D.1 to present some pieces of advice to determine the filtration order in FireGNN. (vtoA)
- We have added the experiments about the number of rewiring used by each algorithm to indicate that our proposed algorithm achieves a better resilience gain than baselines under the same rewiring budget. (vtoA & Qtac)
- We have made it clear that the regular combination of GNN and RL cannot learn to rewire edges to improve network resilience to show the importance of our proposed FireGNN. (Qtac)
- We have moved the related work on using GNN for CO problems from the appendix to Section 2 to describe the related methods that inspire our work. (Qtac)
- We have added the running time of different algorithms to show that our proposed method is more efficient when directly testing on large datasets in Table 4 and Table 5. (Qtac)
- We have added more visualizations of the optimized graphs structures to show that our proposed method can be applied to improve different resilience metrics in Figure 7. (AvkM)
- We have made it clear that studying the resilience task defined on the graph structure is meaningful and well-motivated in the introduction (AvkM)

---

### Decision · Program_Chairs · 2022-01-20

**Decision:**

Reject

**Comment:**

The paper proposes an RL technique for dealing with the problem of network (graph) rewiring for robustness against attacks. Graph rewiring has been studied in a variety of fields, including graph theory (graph abstraction), graph ML (adversarial robustness, performance of GNNs), and combinatorial optimization. Reviewers had concerns with novelty, the correctness of some of the statements, and empirical evalution (in particular, baselines and scalability). While the rebuttal addressed some of the concerns, the overall feel about the paper is lukewarm and the AC believes the paper is below the bar.